# Comparison of land-surface humidity between observations and CMIP5 models

Robert J. H. Dunn[1], Kate M. Willett[1], Andrew Ciavarella[1], and Peter A. Stott[1]

[1]Met Office Hadley Centre, FitzRoy Road, Exeter, EX1 3PB, UK

*Correspondence to:* robert.dunn@metoffice.gov.uk

**Abstract.** We compare the latest observational land-surface humidity dataset, HadISDH, with the latest generation of climate models extracted from the CMIP5 archive and the ERA-Interim reanalysis over the period 1973 to present. The globally averaged behaviour of HadISDH and ERA-Interim are very similar in both humidity measures and air temperature, at decadal and interannual timescales.

The global average relative humidity shows a gradual increase from 1973 to 2000, followed by a steep decline in recent years. The observed specific humidity shows a steady increase in the global average during the early period but in the later period it remains approximately constant. None of the CMIP5 models or experiments capture the observed behaviour of the relative or specific humidity over the entire study period. When using an atmosphere-only model, driven by observed sea-surface temperatures and radiative forcing changes, the behaviour of regional average temperature and specific humidity are better captured, but there is little improvement in the relative humidity.

Comparing the observed and *historical* model climatologies show that the models are generally cooler everywhere, are drier and less saturated in the tropics and extra tropics, and have comparable moisture levels but are more saturated in the high latitudes. The spatial pattern of linear trends are relatively similar between the models and HadISDH for temperature and specific humidity, but there are large differences for relative humidity, with less moistening shown in the models over the Tropics, and very little at high latitudes. The observed drying in mid-latitudes is present at a much lower magnitudes in the CMIP5 models. Relationships between temperature and humidity anomalies ($T - q$ and $T - rh$) show good agreement for specific humidity between models and observations, and between the models themselves, but much poorer for relative humidity. The $T - q$ correlation from the models is more steeply positive in all regions than the observations, and this over-correlation may be due to missing processes in the models.

The observed temporal behaviour appears to be a robust climate feature rather than observational error. It has been previously documented and is theoretically consistent with faster warming rates over land compared to oceans. Thus, the poor replication in the models, especially in the atmosphere only model, leads to questions over future projections of impacts related to changes in surface relative humidity. It also precludes any formal detection and attribution assessment.

# 1 Introduction

Water vapour is a primary greenhouse gas in the atmosphere, modifying the radiation budget and augmenting climate change. Surface humidity is also a source of available water for precipitation, where it governs the amount of rainfall during heavy events where a large fraction of the water is rained out (Trenberth, 1999). Also, energy absorbed during evaporation can be transported and released elsewhere during condensation, redistributing incident solar energy across the globe. Hence humidity, especially at the surface, plays a key role in both the energy and hydrological cycles of the climate system, and is therefore an Essential Climate Variable (Bojinski et al., 2014). Understanding its behaviour over the recent past, and being able to reconstruct this behaviour with Global Climate Models (GCMs) is clearly important for providing robust future projections.

There are a number of ways of characterising the amount of water vapour present in the air (see Willett et al. 2014a for examples), but in this report we will focus on relative and specific humidity. The relative humidity is the amount of water present expressed as a fraction of the amount that would be present if the air were saturated. Specific humidity is the amount of water (in $g$) present per $kg$ of moist air. Hence, changes in this quantity have a direct impact on the amount of precipitable water available.

Near surface relative humidity is an important measure in the thermal comfort of animals, including ourselves, who rely on the evaporation of water (sweating or panting) to thermo-regulate. High relative humidity at high temperatures limits the rate at which evaporation can occur, possibly leading to fatal over-heating in extreme circumstances. At low temperatures, however, moister air can make the body feel cooler through more efficient conduction of heat away from the skin and greater amounts of energy required to warm the moist air close to the skin. Hence changes in relative humidity are important to the health and productivity (e.g. physical work, milk yields etc) of a wide range of fauna.

Since the 1970s, when most humidity monitoring records begin, until the turn of the century, specific humidity has increased over most of the well-observed parts of the globe (Dai, 2006; Berry and Kent, 2011; Willett et al., 2013). This has been largely driven by rising surface temperatures, that have, in turn, increased the water holding capacity of the atmosphere. Where there are few limitations on the amount of water available, the amount of water vapour has increased, largely following the Clausius-Clapeyron relationship (Held and Soden, 2006). The relative humidity appears to have gently risen over that time, albeit with large variability and large observational uncertainty, leading to low confidence in this conclusion. However, since the turn of the century a decrease in the relative humidity and a plateauing of the specific humidity have been observed over land (Simmons et al., 2010; Willett et al., 2014a). Currently, no marine relative humidity dataset exists aside from reanalysis products. Observed marine specific humidity shows a similar pattern to that over land, although the flattening in the 21[st] century is less clear, especially given the El Niño enhanced peak in 1998. Relative humidity over oceans from reanalyses appears approximately constant over both periods (Willett et al., 2015, 2016), though there are indications for a slight decline in dewpoint depresession in the ERA-20CM experiements (Hersbach et al., 2015). Furthermore, slight changes over time in the difference between marine air temperature and sea-surface temperatures (SSTs) in ERA-Interim and also the JRA-55 reanalyses (Simmons et al., 2016), as well as the CMIP5 model archive (Cowtan et al., 2015), suggest that there may be small shifts in the relative humidity of near-surface air over the oceans, but which is not detected in the current reanalyses.

The observed behaviour of specific and relative humidity since the end of the twentieth century has been largely unexpected. Earlier work concluded that relative humidity had remained broadly constant over the 1973-2003 period (Dai, 2006; Willett et al., 2008, 2010) and the expectation was that it would continue to do so in the near term. The older generation Coupled Model Intercomparison Project phase 3 (CMIP3) models, using the Climate of the 20th Century forcing, were found to be in good agreement with the observed global, tropical and Northern Hemisphere average specific humidity changes for the period 1973-1999. However, agreement was poor over the Southern Hemisphere where model trends were positive compared to no trend in the observations. The temperature related change in specific humidity was also in very poor agreement in the Southern Hemisphere. The modelled rate of change was far higher at $\sim 5.5 \% \ K^{-1}$ compared to the observed rate of $-0.27 \% \ K^{-1}$. Although seasonal climatological biases compared to the observations were prevalent in CMIP3 models, no overall tendency towards being overly moist or dry was found.

To reliably project potential future humidity changes, the performance of the latest climate models needs to be assessed against the updated observations to ensure that they are fit for purpose. With the recent completion of the World Climate Research Programme's (WCRP) Coupled Model Intercomparison Project phase 5 (CMIP5, Taylor et al. 2012), and the creation of HadISDH (Willett et al., 2014a), an annually updated humidity monitoring dataset, this is now possible. Future changes in atmospheric water vapour are important to quantify correctly because of the role that water vapour plays in energy transport, the hydrological cycle and also radiative transfer through the formation of clouds. There are significant societal implications from changes in the intensity of heavy downpour events (Kendon et al., 2014), along with those outlined earlier. Ultimately, a better understanding of how humidity (both relatively and in absolute terms) will change in the future is important for good adaptation and mitigation strategy.

Previous studies comparing observations of humidity variables to coupled climate models (CMIP3 or CMIP5) have shown that the observed rise in specific humidity over 1973 to 2003 (Willett et al., 2007; Barkhordarian et al., 2012) and in marine total column water vapour over 1988 to 2006 (Santer et al., 2007) are attributable mainly to human influences. However, large changes in the behaviour of the global average specific and relative humidity in the last decade (Simmons et al., 2010; Willett et al., 2014a, 2016) mean that those studies should be revisited.

This study explores the similarities and differences between the most recent suite of CMIP5 models and the latest observational data, along with a reanalysis product and an atmosphere-only ensemble from a single model. It is a necessary first step in the process of reassessing recent behaviour of global and regional specific and relative humidity.

The observational, reanalysis and model data sources used in this analysis are described in Sect. 2. There are many different ways to compare the models and observations. Herein, we first study local grid-box scale trends in HadISDH to pull out regions where there have been strong or weak changes over the period of the dataset (Sect. 3). We then assess whether the models broadly capture observed features in temperature, specific humidity and relative humidity on the largest temporal and spatial (zonal) scales (Sect. 4). We also assess the differences between *historical* and *historicalNat* forcings to demonstrate whether any signals can be detected and attributed in the loosest sense (*historical* forcings of climate models include anthropogenic and natural factors, whereas *historicalNat* only includes natural factors - for more details see Section 2.2). Thereafter we investigate the spatial detail of both climatology and trends for the three variables to assess whether similarities or differences

are underpinned by similar climatological characteristics in the first place (Sect. 5). Then we explore the strength and slope of temperature-humidity relationships for different regions. In combination with the assessment of background climatology and trends this may reveal whether there are any notable differences in the underlying model physics compared to the observations (Sect. 6). Finally all threads from this investigation are drawn together and discussed in Sect. 7 and summarised in Sect. 8.

## 2  Data Preparation

### 2.1  Observations: HadISDH

HadISDH is a multi-variable humidity monitoring product from the Met Office Hadley Centre (Willett et al., 2014a) expressly designed for the monitoring of surface atmospheric humidity [1]. HadISDH includes a simultaneously observed temperature product (Willett et al, in prep) which is included here for direct comparison with humidity variables. HadISDH is based on the sub-daily, quality-controlled station dataset HadISD which provides temperature and dewpoint temperature data amongst other meteorological variables from 1973 onwards (Dunn et al., 2012). For the detailed methods used to create HadISDH from HadISD see Willett et al. (2014a), but we reproduce an outline below. A subset of the 6103 HadISD stations were selected on the basis of their length of record and data quality, leaving around 3500 stations (the exact number is dependent on the humidity variable). The sub-daily data from each of these stations were converted to monthly mean values and subsequently homogenised to remove non-climatic features. The Pairwise Homogenisation Algorithm (Menne and Williams Jr, 2009) was used for this step, but to ensure consistency across the different humidity variables an indirect approach was applied using the change point locations derived from the dew point depression and temperature values (see Willett et al., 2014a for details).

After homogenisation, the adjusted station monthly means were gridded onto a $5° \times 5°$ grid by simple averaging. HadISDH suffers from the spatio-temporal coverage issues common to most *in situ* global data products; station density is generally sparse outside of the USA and Europe, especially for high latitudes, tropics, South America, Africa, the Middle East and most of Australia. Furthermore, despite extensive efforts to quality control and homogenise it is highly likely that some errors remain. To account for this uncertainty estimates arising from the station (measurement, climatology, inhomogeneity) and gridding (spatio-temporal coverage) have also been calculated for each grid box and time. These uncertainties have been included when calculating large-scale average time series along with that arising from incomplete coverage across the globe.

In this analysis we use monthly mean anomalies relative to the 1976 to 2005 period from HadISDH version 2.0.1.2015p (unless otherwise stated). Global and regional monthly means were calculated for each variable using cosine-weighting of the grid-box latitude; with annual means calculated from the monthly means.

### 2.2  CMIP5 models

The CMIP5 has provided a valuable repository for a wide range of climate model data (Taylor et al., 2012). Over 60 climate models with numerous experiments each have been provided for use by over 20 different institutions. In this study we focus on

---

[1]HadISDH is available at www.metoffice.gov.uk/hadobs/hadisdh/.

| Model | historical | historicalExt | historicalNat | historicalGHG | SI | CA | LU | O3 |
|---|---|---|---|---|---|---|---|---|
| bcc-csm1-1[*] | 3 | - | 1 | 1 | N | Y | N | N |
| CanESM2 | 5 | 5 | 5 | 5 | Y | Y | Y | N |
| CNRM-CM5 | 10 | 10 | 6 | 6 | Y | Y | N | Y |
| CSIRO Mk-3-6-0 | 10 | - | 5 | 5[‡] | Y | Y | N | N |
| GISS E2-H | 6 | 6 | 5 | 5 | Y | Y | Y | N |
| GISS E2-R | 6 | 5 | 5 | 5 | Y | Y | Y | N |
| HadGEM2-ES | 4 | 3 | 4 | 4 | Y | Y | Y | N |
| IPSL CM5A-LR[‡] | 6 | - | 3 | 3 | Y | Y | Y | Y |
| NorESM1-M | 3 | 3 | 1 | 1 | Y | Y | N | Y |

**Table 1. CMIP5 models**. The number of ensemble members for each model and experiment. We did not use ensemble members derived from perturbed physics simulations. All models include volcanic aerosol influences. SI - sulphate indirect effects (first and/or second effects), CA - carbonaceous aerosols (black and organic carbon), LU - anthropogenic land use changes in the *historical* experiment, O3 - ozone influences in the *historicalGHG* experiment. [*] *historical* experiment runs until 12-2012 without separate *historicalExt* experiment. [†] There are no ensemble members for the specific humidity. [‡] Volcanic stratospheric aerosols included by varying solar irradiance and land use changes also included with *historicalGHG* experiments.

the set of simulations of the 20th century (1850-2005) with different forcing factors applied. The "historical" experiments are forced by anthropogenic and natural factors to capture as closely as possible the recent climate. Extra simulations have been run for some models to extend the historical period to 2012 or 2014 ("*historicalExt*"). There also exist simulations forced by only natural factors ("*historicalNat*") and those forced only by greenhouse gas factors ("*historicalGHG*"). Not all models have simulations of these last two experiments run beyond 2005. To study the changes in surface humidity in the CMIP5 archive, especially over the last decade, we select the nine models which have coverage up to at least 2012 in their *historicalGHG* and *historicalNat* experiments (see Table 1). The *historicalExt* experiments were merged with the *historical* runs where possible to provide coverage beyond 2005, though for three models no *historicalExt* experiments exist. For a more comprehensive description of the different models and their individual forcing factors relevant to this analysis we refer to Section 2 of Jones et al. (2013).

A number of specific features in each of the models could strongly influence their ability to capture changes in humidity measures. Land surface processes such as changes in soil moisture or $CO_2$ fertilisation effects on evapotranspiration can strongly influence the humidity measures. Replication of realistic aerosol forcings, and particularly the date, location and magnitude of volcanic forcings is also important. Hence the degree to which models include these effects will impact the degree to which they replicate observed humidity changes.

Anomalies for each experiment run were calculated relative to the 1976 to 2005 period to match HadISDH. As HadISDH is a land-surface only dataset and has varying coverage with time, this needs to be accounted for in the model coverage when creating global and regional averages. Each month in the CMIP5 models was regridded to the $5° \times 5°$ resolution of HadISDH,

and then coverage matched with the corresponding month in HadISDH. Global and regional monthly means were calculated for each ensemble member using cosine-weighting of the grid-box latitude; with annual means calculated from the monthly means.

## 2.3 Atmosphere-only HadGEM3

The coupled models from the CMIP5 archive are driven by long-term radiative forcing parameters (including volcanic aerosols, solar activity, greenhouse gas emissions etc). This ensures that they should capture long-term trends and changes. However, they are not expected to match the temporal pattern of short-term variations driven by modes of variability such as the El Niño Southern Oscillation, for example. Despite this, they are expected to capture the amplitude of these variations successfully.

Atmosphere-only models use observed sea-surface temperatures (SSTs) and radiative forcings to drive the atmospheric
portion of a coupled-climate model. This additional constraint should ensure that these models more accurately capture the short-term variations of the observed climate at the same point in time. Furthermore, if there are large scale changes in the SSTs that are not captured by the coupled models, then an atmosphere-only model should improve the match to the observations.

The land surface has been warming faster than the oceans over the last 10 to 15 years, a characteristic that has not been well captured by coupled climate models. By including a model driven by observed SSTs, we can assess to what extent any
land-ocean heating contrast is driving any differences between the modelled and observed humidity measures (see also Section 7). Also, the representation of large scale atmosphere-ocean circulation patterns should be improved in an atmosphere-only model, and hence short timescale variations in the temperature and humidity may be as well.

We use the latest version of the Hadley Centre model in its atmosphere-only configuration, HadGEM3-A (Walters et al., 2011; Hewitt et al., 2011; Ciavarella et al., 2017), to compare with the coupled version of its predecessor, HadGEM2-ES. We
use an ensemble of 15 equivalent realisations initialised in December 1959 and run under historical forcings, consistent with the CMIP5 generation of coupled models, and at relatively high resolution of N216 L85 ( 60km midlatitudes). This has also been processed to match the HadISDH data in the same way as the CMIP5 models.

## 2.4 Reanalysis data: ERA-Interim

Willett et al. (2014a, 2015, 2016) show that HadISDH is in broad agreement with the annual time series from a number of
25 reanalyses datasets. Of the most recent products, only ERA-Interim (Dee et al., 2011) and JRA-55 (Kobayashi et al., 2015) provide direct analysis of 2 m air temperature and humidity. Willett et al. (2016) show that there is better agreement between these two products than betwen either of them and the MERRA-2 reanalysis (Bosilovich et al., 2015; Gelaro et al., 2017), especially for relative humidity. MERRA-2 also shows inconsistencies to other reanalysis products in surface air temperature (Sanchez-Lugo et al., 2016).
**Do we include JRA-55 - would need to update many of the plots, but is it worth it for completeness**

We include the ERA-Interim reanalysis (Dee et al., 2011) dataset in our assessment as it is in very good agreement with a range of observational products (Simmons et al., 2010; Willett et al., 2014b). The specific and relative humidity fields were calculated from the 6-hourly fields of temperature, dewpoint temperature and pressure. These sub-daily fields were then

averaged to monthly values and the data were re-gridded to the $5° \times 5°$ grid of HadISDH. Then each month was coverage matched as for the CMIP5 models with identical calculations to obtain the global and regional averages. The climatology period was 1979-2005 to mirror the HadISDH period as closely as possible, as there is no data prior to 1979 in ERA-Interim.

The ERA-Interim reanalysis product, although having the advantages of complete coverage and using physics-based algorithms to calculate the humidity parameters, has a some limitations in terms of long-term stability. A key example is the change in input SSTs. From 1979-1981 the Met Office Hadley Centre monthly HadISST1 was used. This was then swapped to the US National Centers for Environmental Prediction (NCEP) weekly 2Dvar dataset and then again in June 2001 to the daily operational NCEP product. A further change occurred in January 2002, which combined with the June 2001 change resulted in a shift to lower SSTs of approximately 0.15K globally, which is now accounted for by some studies (e.g. Simmons et al., 2016). In February 2009 the source shifted again to OSTIA, which has more sea ice and higher SSTs at the poles, but the differences were mostly negligible (Simmons et al., 2010; Dee et al., 2011; Simmons and Poli, 2015). Over time there have been various changes to the types and density of observing platforms available. While the assimilation system can mitigate this to some extent some inhomogeneities occur.

## 3   Regional and local grid-box scale trends in HadISDH

To give overall context for the later sections which compare the behaviour of the CMIP5 models to the HadISDH and ERA-Interim in the temporal and spatial domains as well as in temperature-humidity relationships, we begin with a quick overview of humidity in HadISDH and ERA-Interim.

The behaviour of the surface air temperature, the specific and the relative humidities in HadISDH is shown in Fig. 1 as linear trends calculated over each of four decades (1973-82, 1983-92, 1993-2002, 2003-15). The trends have been calculated using the median of pairwise slopes estimator (MPW, Theil, 1950; Sen, 1968; Lanzante, 1996). Different parts of the globe dominate the warming or moistening signal in each of the four decades. For the surface air temperature, the most intense and widespread warming out of the four panels occurred between 1993 to 2002 (Fig. 1*m*). The specific humidity also had the strongest moistening during that decade, but the strongest drying occurred between 2003-15. For the relative humidity, the strongest drying occurred in the same decade as for the specific humidity, but the strongest moistening between 1983-92 (Fig. 1*n-o*).

The decrease in the relative humidity found by Simmons et al. (2010) and supported by Willett et al. (2014b) is clear in the Fig 1*l* compared to Fig. 1*c*, *f* and *i*. Strong drying is widespread, especially over the Northern Hemisphere mid-latitudes and particularly in North America and Central and Eastern Asia in the last two decades. For the tropics and Southern Hemisphere, there are changes (moistening in 1993-2002 in southern South America and southern Africa, and drying in South America in 2003-2015), but none are as widespread as in the Northern Hemisphere. Comparing the trends over the entire historical period of record (see Fig. 6), historical drying is less widespread but more zonal, with strong moistening over India. We note that while expected changes in near-surface relative humidity by 2100 under a business-as-usual (RCP 8.5) scenario show widespread reductions, only a few small areas (sub-Saharan Africa, India, parts of Argentina) showing (non-significant)

increases (Sherwood and Fu, 2014, Fig. S1). These projected end-of-century changes align with the observed changes in HadISDH, but appear to occur on much longer timescales.

The specific humidity is more complex, with a moistening observed in the first three decades, but a more mixed picture (which averages out to no change) in the last. However, the areas which show drying are not consistent from decade to decade, with large regions strongly drying in one and moistening in the next (e.g. South Africa in 1983-1992 versus 1993-2002). The areas which moisten do roughly correspond to the areas which experience the largest temperature trend over the same period. Some correspondence is expected, as warmer air can hold more moisture, but this is influenced by large scale circulation patterns and also the availability of water.

## 4   Comparison of large scale temporal evolution between observations and models

We now assess whether the models broadly capture observed features on the largest temporal and spatial scales. The global annual time series for a selection of the CMIP5 models and HadGEM3-A are shown for air temperature, specific humidity and relative humidity as Figs. 2 to 4 respectively. The models shown here are those which have *historicalExt* experiments (see Table 1), but only one of the GISS models is shown. The figures for all models, variables and regions are available in the Supplementary Material (Figs. 1-4, 10-13, 24-27). When more than one ensemble member is available we show the ensemble average for each time step. The ensemble spread is calculated using the method from Tett et al. (2002), by scaling the residuals of the $m$ individual members from the average by $\sqrt{m/(m-1)}$. In each main panel HadISDH (black) with uncertainty range (grey) and ERA-Interim (magenta) are also shown.

In the three small right hand panels of Figs. 2-4 we show the results of a linear trend analysis on HadISDH, ERA-Interim and also the ensemble means of the different model experiments. For this we have again used the MPW method. The colours of the symbols match those of the timeseries in the main panel. The errorbars show the 90[th] percentile range of the slopes determined by the MPW method to give an indication of the confidence range of the calculated slope. The rightmost of these three small panels shows the linear trend over the full span of the data. The other two show the linear trend over the early and late period by splitting the timeseries into 1973-1994 (22 years) and 1995-2015 (21 years). For each panel, the trend for HadISDH is shown both for the full period and the one which matches the model coverage. We note that the ensemble means of the CMIP5 model experiments will have less noise and variation than the individual ensemble members, and hence show smoother changes with smaller confidence limits on the trends. For the late and full period panels, the HadISDH trend is shown matching the temporal coverage of the *historical* model (circle) and its full coverage (cross).

### 4.1   Temperature

While the main focus of this paper is humidity, it is helpful to also look at temperature as one of the key drivers of changes in specific and relative humidity. Several studies have conducted formal detection and attribution of global land surface temperatures, with the best explanation for recent trends requiring the inclusion of greenhouse gas forcing (e.g. Bindoff et al., 2013). The observations show strong warming trends for the early, late and full period (Fig. 2). As expected, the atmosphere-only

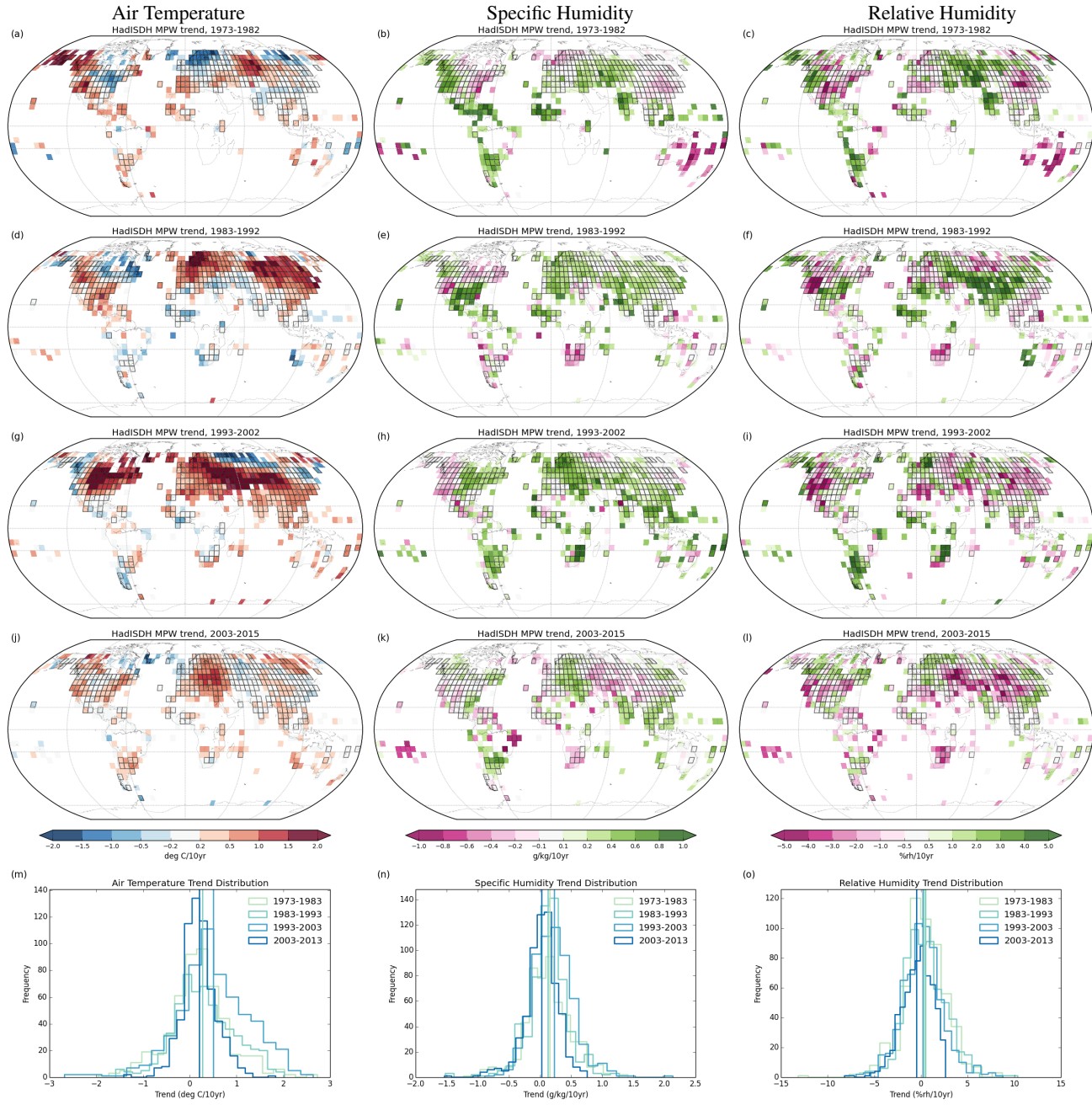

**Figure 1.** Decadal trends (over 13 years for final interval) as calculated from HadISDH for Temperature, specific humidity and relative humidity. Slopes have been calculated using the robust, median of pairwise slopes estimator (MPW, Theil, 1950; Sen, 1968; Lanzante, 1996). Grid boxes which contain a median of three or more stations in each month have been highlighted with a thick line. Bottom three panels (*m - o*) show the distribution of the trends from each of the four periods. The vertical lines show the latitude weighted mean of the trends in each decade.

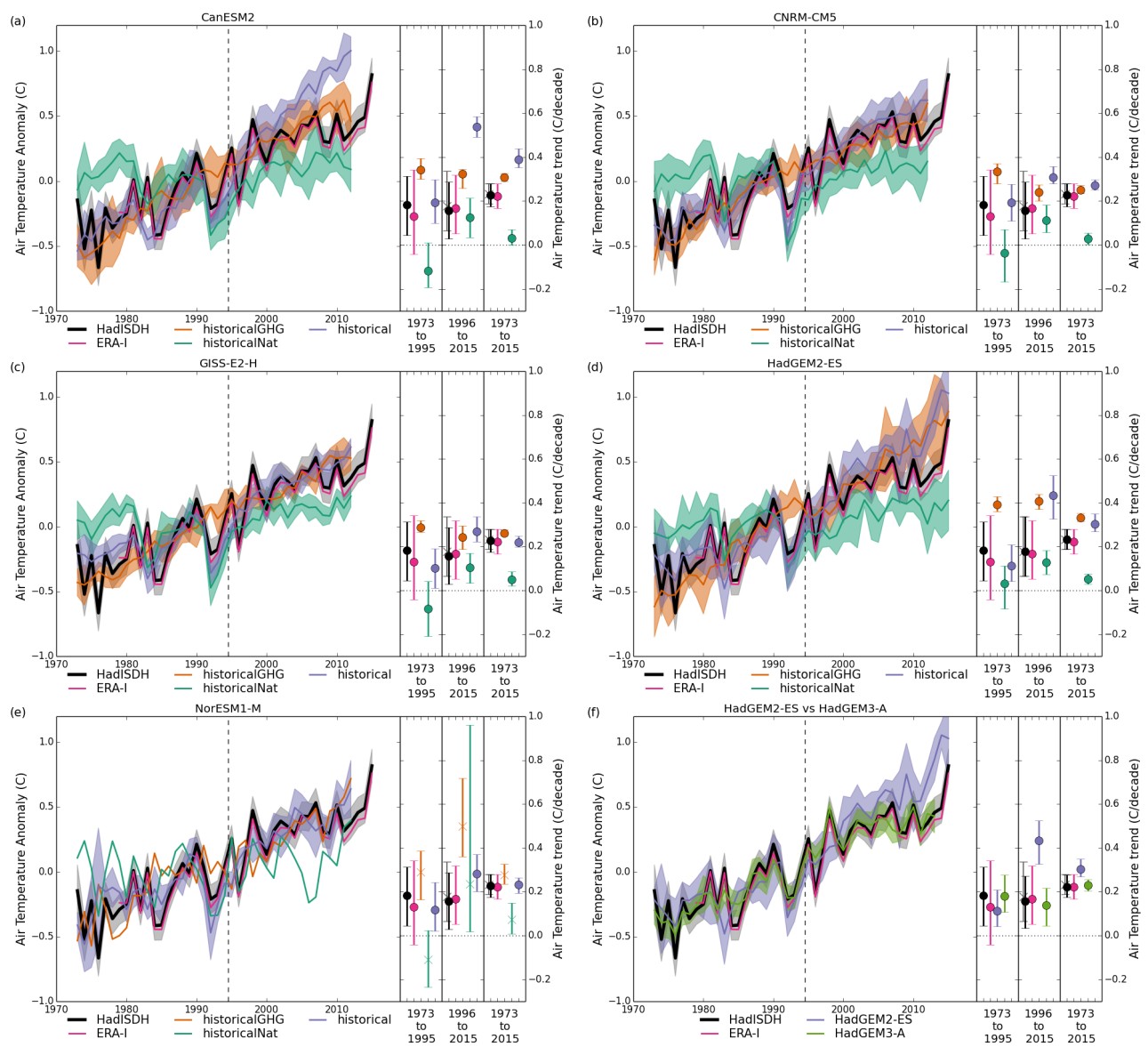

**Figure 2.** Global annual timeseries for temperature for five of the nine CMIP5 models and HadGEM3-A, using a climatology period of 1976-2005. HadISDH is shown by the thick black line, ERA-Interim by the magenta line and the historical, historicalNat and historicalGHG ensemble averages by the purple, green and orange lines respectively. The uncertainty ranges are shown using the coloured shading. The right-hand panels show the values of the linear trends for HadISDH, ERA-Interim and the ensemble averages of all three experiments for the early (1973-1994), late (1995-2015) and full periods. For the late and full period panels, the HadISDH trend is shown matching the temporal coverage of the *historical* model (circle) and its full coverage (cross). If there is only one ensemble member for the model, then the trend is marked with a cross rather than circle

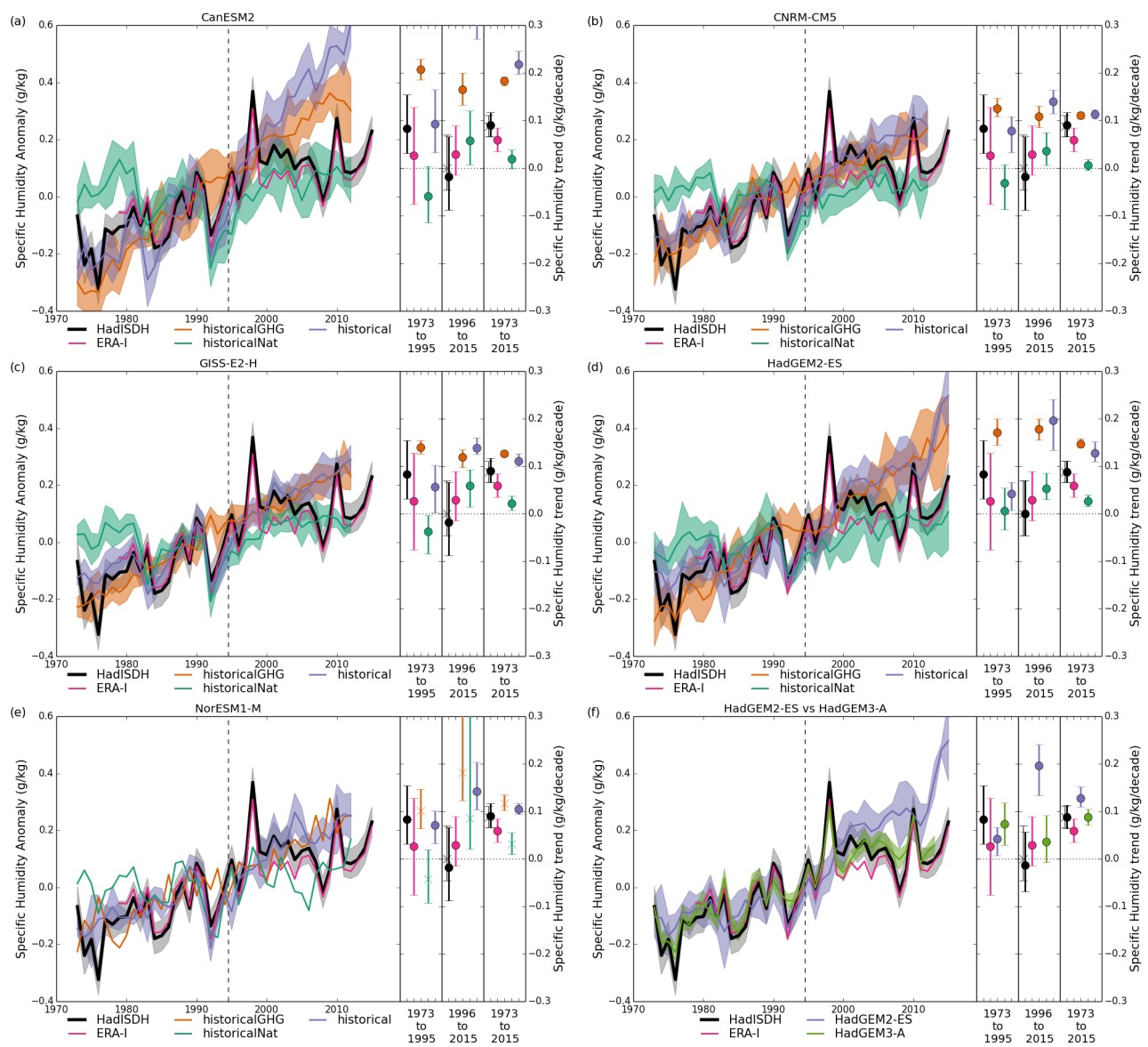

**Figure 3.** Global annual timeseries for specific humidity for five of the nine CMIP5 models and HadGEM3-A, using a climatology period of 1976-2005. HadISDH is shown by the thick black line, ERA-Interim by the magenta line and the historical, historicalNat and historicalGHG ensemble averages by the purple, green and orange lines respectively. The uncertainty ranges are shown using the coloured shading. The right-hand panels show the values of the linear trends for HadISDH, ERA-Interim and the ensemble averages of all three experiments for the early (1973-1994), late (1995-2015) and full periods. For the late and full period panels, the HadISDH trend is shown matching the temporal coverage of the *historical* model (circle) and its full coverage (cross). If there is only one ensemble member for the model, then the trend is marked with a cross rather than circle.

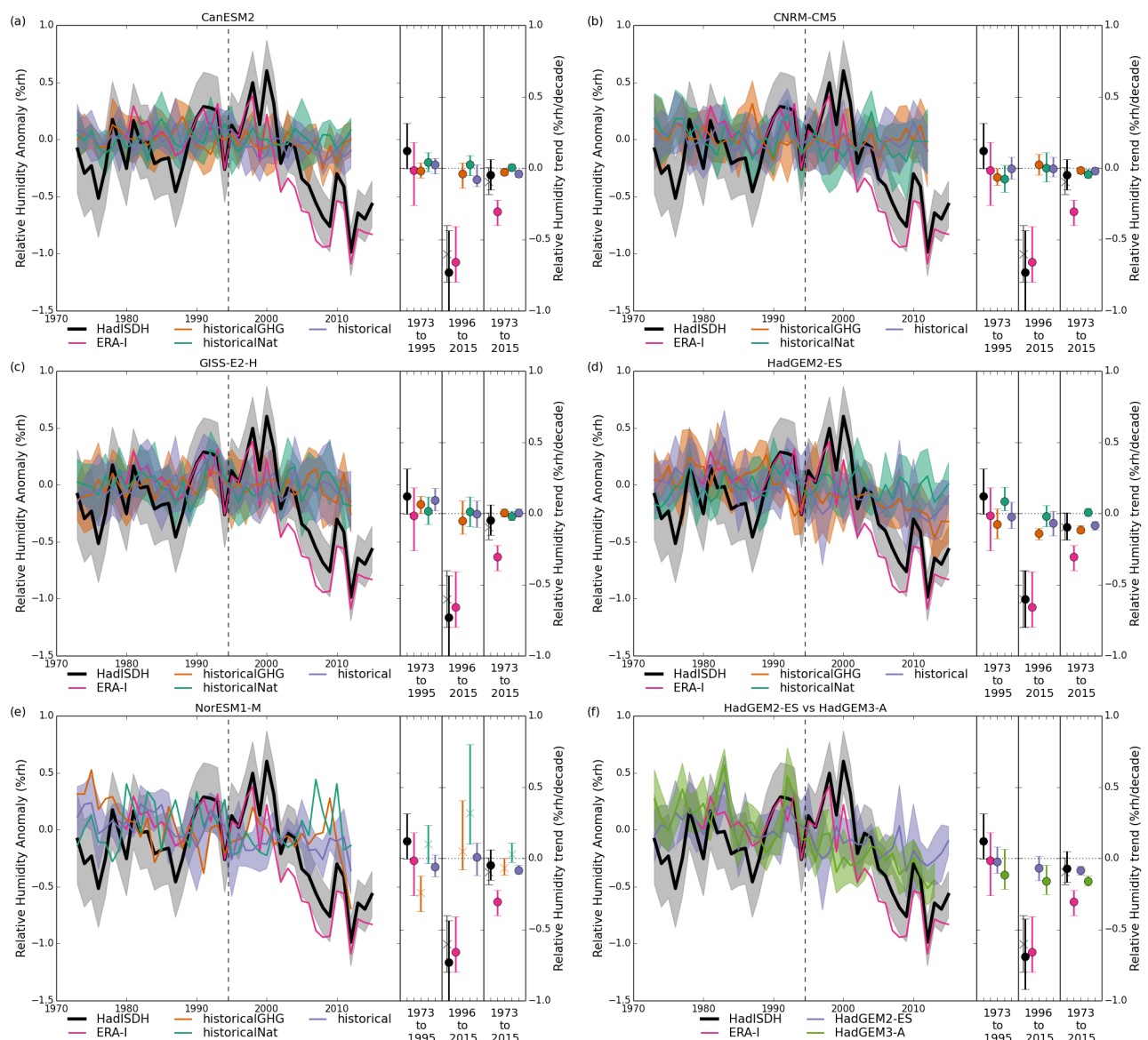

**Figure 4.** Global annual timeseries for relative humidity for five of the nine CMIP5 models and HadGEM3-A, using a climatology period of 1976-2005. HadISDH is shown by the thick black line, ERA-Interim by the magenta line and the historical, historicalNat and historicalGHG ensemble averages by the purple, green and orange lines respectively. The uncertainty ranges are shown using the coloured shading. The right-hand panels show the values of the linear trends for HadISDH, ERA-Interim and the ensemble averages of all three experiments for the early (1973-1994), late (1995-2015) and full periods. For the late and full period panels, the HadISDH trend is shown matching the temporal coverage of the *historical* model (circle) and its full coverage (cross). If there is only one ensemble member for the model, then the trend is marked with a cross rather than circle.

HadGEM3-A model shows the best agreement with observed estimates, whereas the *historical* runs from the CMIP5 models all have larger positive trends than the observations for the later period (see Section 7 for discussion of the most recent years). Most of the CMIP5 *historical* ensemble means are consistent with the observations for the Globe, Northern Hemisphere and Tropics, in that the confidence ranges of linear trends overlap over the entire period of study (see Supplementary Information Figs. 1-3, rightmost trend panels). For the Southern Hemisphere however, the majority CMIP5 *historical* models have trends that are much more positive than the observed trends with no overlap of confidence ranges (Supplementary Information Fig 4, rightmost trend panels). Note that nearly all of CMIP5 *historicalNAT* ensemble mean trends over the full period, although positive, are much smaller and inconsistent (the spread does not overlap) with the *historical* trends (all regions) and the observed trends (all regions except for the single member *historicalNAT* runs in NorESM1-M and bcc-csm1-1 in the tropics and these plus IPSL-CM5A-LR in the Southern Hemisphere). In the Southern Hemisphere, the observed trends are more consistent with the *historicalNat* trends than the *historical* trends. This suggests that although the CMIP5 models show clear anthropogenically induced changes over all regions which are larger than the observed trends, the contribution of GHGs could be part of the explanation for the observed trends, apart from in the Southern Hemisphere. Jones et al. (2013) show in their Fig. 8, that compared to both *historical* and *historicalGHG* experiments, both the observations (HadCRUT4, Morice et al., 2012) and the *historicalNat* experiment show cooling over the most recent three decades. Given the dominance of the oceans in the Southern Hemisphere, this may influence the land-based observations in HadISDH.

## 4.2 Specific Humidity

Specific humidity in HadISDH exhibits a positive moistening trend in the early period for all regions bar the Southern Hemisphere, with short-term decreases corresponding to the two significant volcanic eruptions in this period – El Chichon[2] and Pinatubo (Fig. 3). There is no clear trend in the later period in any of the regions. The agreement between the models and observations is worse for specific humidity and relative humidity than for temperature, as Figs. 3 and 4 show. For specific humidity, the CMIP5 model *historical* ensemble spreads still broadly show consistency with the observed full-period moistening trends, in that they are all positive and the confidence ranges overlap, except for the Southern Hemisphere where the observed trends are essentially zero (Supplementary Information Fig. 27) and CanESM2. However, like temperature, the CMIP5 *historical* full period trends are consistently larger than the observed trends, especially in the Southern Hemisphere. Note that the CMIP5 *historical* trends are inconsistent with *historicalNAT* trends for all regions. Hence, as for temperature, this suggests that the action of GHGs may be part of the explanation for the observed trends, apart from in the Southern Hemisphere. However, analysis of the time series behaviour, and differing trend directions between models and observations in the two shorter periods (especially apparent in the late period) reveal that even with greenhouse gas forcings, the interdecadal agreement is poor. This suggests that in recent years, changes in atmospheric greenhouse gases may have had limited impact on land surface specific humidity. Also, as the CMIP5 models have not followed the slower warming rate over the last two decades, we cannot expect them to have followed the stalling in the rate of moistening.

---

[2]The eruption of El Chichon in 1982 has an apparent delayed effect on the specific humidity because of the 1982-83 El Niño which canceled out some of the decrease.

The agreement between specific humidity in the atmosphere-only HadGEM3-A and the observations is far better, at all timescales and especially in the later period, similar to the results for temperature. Given the more positive warming trends in the CMIP5, we would expect larger trends in specific humidity on these large scales. As the warming in HadGEM3-A is constrained to some extent by use of observed SSTs, which in turn leads to the same observed temporal evolution of modes of natural variability (e.g. El Niño-Southern Oscillation (ENSO), North Atlantic Oscillation (NAO)), this closer agreement is to be expected. Interannual variability is generally in good agreement between HadISDH (and ERA-Interim) and HadGEM3-A, better than for the CMIP5 coupled models. However, there are quite large differences in the Southern Hemisphere, especially in more recent years, corresponding with the La Niñas of 2008 and 2011-12. Interestingly, the interannual agreement appears much better for temperature than for specific humidity (Supplementary Information Figs. 4 and 27).

While large scale features are still consistent over the current 40 year period, given the inconsistency in the most recent period, it is clear that a few more years of data could change that. This has implications for any detection and attribution studies on specific humidity and also studies using models to look at future changes in specific humidity.

### 4.3 Relative Humidity

The observed relative humidity exhibits negative trends over the full period, although the early period shows positive trends for the Globe and Northern Hemisphere, clearly inconsistent with no trend in the case of the latter region (Fig. 4). In contrast the Northern Hemisphere ERA-Interim does not show strong positive trends in the early period. The Northern Hemisphere dominates the global signal, as is to be expected given the larger spatial coverage over this region. Here the full period trends mask very different behaviour in the early and late periods, even when the temporal coverage of the CMIP5 *historical* models are taken into account.

For relative humidity, there is far greater spread across the models and different forcings than for temperature and specific humidity, and in general, agreement with the observations is much poorer, even for HadGEM3-A. Although some models show a small positive trend in the early period (GISS-E2-H and GISS-E2-R; Supplementary Information Fig. 10), none show such a strong negative trend for the late period. Generally, all models exhibit relative humidity trends that are closer to zero (neutral trend) than in the observed estimates, with the majority showing small negative trends, except over the Tropics.

For all regions apart from the Southern Hemisphere, the confidence range of the full period observed trends reaches or crosses the zero line, so strictly speaking, although all observed trends are negative, they are only considered "significant" for the Globe and Southern Hemisphere for some models. Over the Northern Hemisphere and Globe the CMIP5 full period *historical* trends broadly agree with the observed negative trends. Over the Southern Hemisphere, where full period observed trends are negative and largest, there is little agreement with the models. When partitioning the observed period into the early and late sections, then it is clear that none of the the CMIP5 models show the strong decline in relative humidity observed in the later period in all regions regardless of the temporal coverage used for the observations. The HadGEM3-A averages have a slightly more negative trend than the CMIP5 models, but it is a small improvement to the match with observations. A similar result was noted by Hersbach et al. (2015) in the ERA-20CM experiments, where the driest conditions were found in the final decade of the 1901-2010 temporal coverage.

The relative humidity timeseries show larger interannual variability than specific humidity or temperature, especially compared to the magnitude of the changes observed in the latter period compared to the earlier. This is partly due to relative humidity being more sensitive, reflecting both changes in temperature and changes in dewpoint temperature directly, whereas specific humidity is only directly affected by changes in the dewpoint temperature. Longer term increases in temperature do drive changes in specific humidity but these are energetic rather than direct, hence the increased sensitivity of relative humidity measures. The large interannual variability of relative humidity means the linear trends have much larger confidence ranges. This results in broad scale consistency between the CMIP5 model *historical* ensemble spread and the observations; most but not all of the full period *historical* trends and *historicalNat* trends overlap the observations over almost all regions, though this is less true for the Southern Hemisphere. However, the interannual and interdecadal variability and the short period trends really are quite different. There is no consistent difference between CMIP5 *historical* and *historicalNat* full period trends in that their confidence ranges overlap. Only for the Northern Hemisphere and Globe are the *historical* trends mostly more negative than the *historicalNat* trends. Hence, unlike temperature and specific humidity, there is no clear GHG driven signal in the relative humidity in the CMIP5 models.

## 4.4 Summary of Temporal Behaviour

For large scale averages, over the current period of record, CMIP5 *historical* models are in broad agreement with the observed long term trends for temperature and specific humidity, but not for relative humidity. The CMIP5 models warm and moisten too much in all regions and do not decline in saturation enough in any region, nor do they agree on whether relative humidity should or should not decline over this period. The best explanation for all regions, except the Southern Hemisphere, are when the models include greenhouse gases. Curiously, the models show strong greenhouse gas driven changes in temperature and specific humidity in the Southern Hemisphere that are not present in the observations.

As noted in Section 4.1, the cooling observed in the Southern Ocean (and also eastern Pacific) may impact the land-based measurements of temperature, but especially humidity. As this cooling is not seen in the *historical* and *historicalGHG* models for surface temperature, then it is unsurprising that the agreement between the observations and *historical* experiments is less good in this region. Jones et al. (2013) note that this cooling may be the result of the Southern Annular Mode (Trenberth et al., 2007) but also suggest that there may be forcing contributions to these changes (Karpechko et al., 2009). A further complication is whether low cloud cover plays a role as this is naturally associated with humidity close to the land surface, but also with aerosols. The same study (Jones et al., 2013) show that models which include indirect aerosol effects have better agreement with the observed global temperature trends. The lack of agreement between the *historical* experiments and observations is also seen in the specific humidity (see Supplementary Information, Fig. 27), as a cooler ocean would result in less moisture in the air blown onto the land. We note that no clear difference between the different experiments is visible for the relative humidity (see Supplementary Information, Fig. 13).

For all variables the behaviour of HadISDH is for the most part mirrored by ERA-Interim, including in all the regional averages. The largest differences are observed in the early 2000s, across both variables and all regions. This indicates that this may be the result of the shift in the source of the SSTs used as input to ERA-Interim (Sect. 2.4).

The agreement in decadal variability and short period trends is actually very poor; best for temperature, less good for specific humidity and worst for relative humidity. This suggests that our conclusion of good agreement in long-term trends may not hold as the record grows year-after-year.

Monthly global timeseries are shown in the Supplementary Material (Figures 5, 14 and 28). The long-term trends calculated in from these timeseries are very similar in magnitude as to the ones from the annual timeseries. Both the models and observations show monthly variability in the temperature and specific humidity, though the magnitudes differ. However, the relative humidity does not show any regular variation on monthly timescales.

The behaviour of the observed and modelled humidity since 1973 suggests that water availability is less of a limiting factor in the models than in the observations. Even though land surface temperatures have warmed more in the models than observed in HadISDH, there appears to still be enough moisture available over land in the models to increase specific humidity at a rate where the relative humidity remains close to constant, as also clear in Fig. 3. From the observed HadISDH, limits on the water availability both over land and in terms of moisture advected from the oceans, combined with increasing land temperatures have been discussed as possible drivers of a plateau in the specific humidity, and a decline in the relative humidity in recent decades (Simmons et al., 2010; Joshi et al., 2008).

Even when modes of variability (key drivers of change, e.g. ENSO via SSTs) are aligned, and land-temperature trends are similar (e.g. by using HadGEM3-A), the trend for specific humidity is still too large, and the agreement for relative humidity is only slightly better than for the coupled models of the CMIP5 archive (Supplementary Information Fig. 10). This suggests that specific and relative humidity do not just depend on natural variability or the amount of warming, something else is missing. In the remaining sections of this manuscript we test a number of possible ideas including different background climatology (Sect. 5.1), different spatial patterns of change (Sect. 5.2), or that the water limitations are less for models than observations impacting the strength and shape of temperature-humidity relationships (Sect. 6). Observational and model errors could also play a part, and there are many other processes that could affect the models ability to match the patterns shown in the observations which may be avenues for further investigation, including evapotranspiration processes (e.g. land-cover type and changes, stomatal conductance/resistance changes under increasing $CO_2$, cloud cover and changes) and differences in the land-sea warming rate.

## 5   Comparison of the spatial pattern of climatology and trends between observations and models

Having assessed the temporal behaviour on global and zonal annual averages, we next assess the similarity of spatial patterns in both the background climatology and long-term linear trends. Climatologically, there are likely to be some regions within the models that are biased relative to the observations. If this is the case it is useful to assess to what degree such features are similar across different models and whether there is any consistency with identified differences in spatial patterns of trends. For this assessment, only the *historical* forced CMIP5 models are compared to HadISDH, along with the atmosphere only HadGEM3-A and the ERA-Interim reanalysis. Summaries of differences are shown in Figs. 5 and 6 for the climatology and long-term trends respectively. Individual model (ensemble mean) minus HadISDH fields are shown in the Supplementary Material (Figs. 7, 9, 16, 18, 30, 32).

## 5.1 Climatological averages

To provide context for the differences between the observations and models/reanalyses, we give a quick description of the climatologies shown in Fig. 5*a* to *c*. The air temperature is unsurprisingly highest in the tropics, and lowest at high latitudes, though large high-altitude regions stand out, e.g the Himalayas. The impact of prevailing westerly flows in the mid-latitudes can be seen on the western coasts of both Europe and North America. The relative humidity clearly shows the less saturated areas over the desert belts around the Tropics of Cancer and Capricorn. In central Asia this also stretches north of the Himalayas on the Tibetan Plateau. High altitude areas in the western US also have lower relative humidities. Coastal areas in the mid-high latitudes along with rainforest regions in South America, western Africa and south-east Asia have higher relative humidities. The specific humidity is highest over the tropics and also coastal regions in the mid-latitudes, e.g. around the Mediterranean Sea. Low values are found in typically arid areas e.g. the deserts of the world, and also the high latitudes.

Panels *d* to *i* in Fig. 5 show the frequency of *historical* CMIP5 models with positive (Fig 5 *d* to *f*) and negative (Fig 5 *g* to *i*) biases relative to the observations; the number of runs across all models and ensemble members which are greater or less than HadISDH at each grid cell. The lower panels (Fig 5 *j* to *o*) show the differences from HadISDH of ERA-Interim and HadGEM3-A.

The *historical* forced CMIP5 models exhibit a diverse range of differences compared to the HadISDH climatology, both between models and also spatially, within any one model. Overall and across all the models, there is a tendency for the CMIP5 models to be too cool over much of the globe (comparing Fig 5 *g* and *d*); aside from coastal eastern USA, parts of South America, mid-central Europe and eastern Japan which appear consistently warmer than HadISDH. The CMIP5 models appear drier and less saturated over the lower latitudes (Fig 5 *h, i*). Conversely, over higher latitudes polewards of $40°$N/S, while specific humidity is broadly similar (Fig 5 *e, h*) the models tend to be too highly saturated (Fig 5 *f, i*). The patterns are more zonal for specific humidity and relative humidity than for temperature. For specific humidity, notably, the largest differences occur in the deep Tropics with very small differences found in the high latitudes. For relative humidity the differences appear large throughout. Most of the CMIP5 models have a cooler bias, are drier especially in the Tropics and are more saturated in the mid-high latitudes. The CanESM2 and NASA-GISS models stand out from the others as having a warm bias. This is very widespread in CanESM2, which also shows a widespread moist and saturated bias (see Supplementary Material Figs. 7, 16 and 30).

Regionally, there are some consistent features across the models. Western and eastern (excluding the central and the east coast) USA, western China, northern Europe and southern South America are cooler and have considerably higher relative humidity (are more saturated) in the historical forced CMIP5 models but with no clear, common differences in specific humidity. So, these regions are more saturated in the models, most likely because they are cooler rather than because they contain less moisture (Fig. 5*g*). Western and northern Africa tend to be cooler with lower specific humidity (less moisture) in the CMIP5 models, but no particularly common difference in relative humidity (saturation). Conversely, the east coast of the USA appears warmer and more moist (higher specific humidity) in the CMIP5 models, again with no particularly common difference in saturation level (relative humidity).

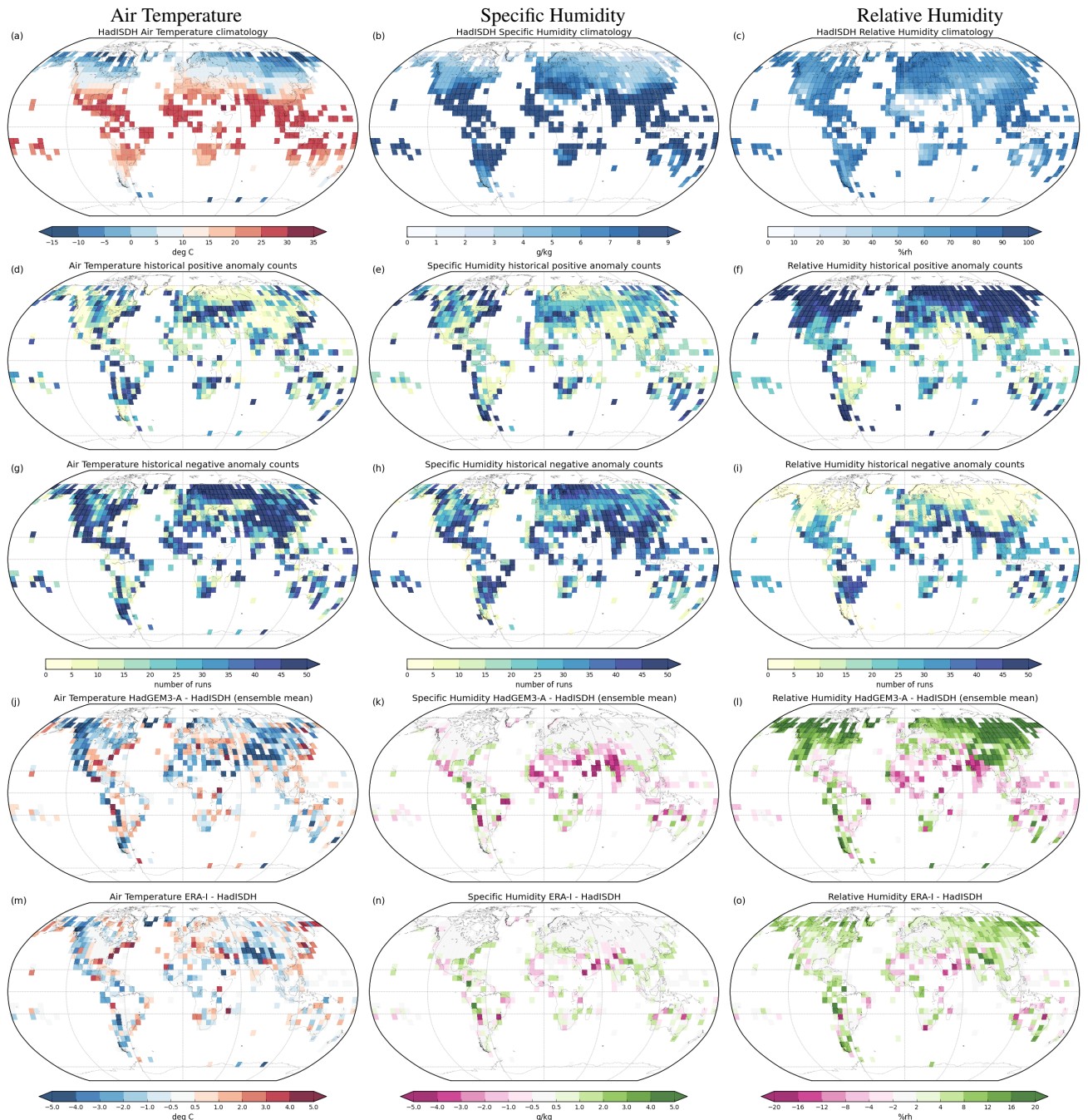

**Figure 5.** Climatological differences between models and observations at the gridbox scale. a, b, c) The climatology of HadISD for air temperature, specific humidity and relative humidity respectively. Frequency of historically forced CMIP5 models with d, e, f) positive bias (warmer/moister/more humid) or g, h, i) negative bias (cooler/drier/more arid) relative to HadISDH for air temperature, specific humidity and relative humidity respectively. For d) to i) shading scales with the multi-model mean difference of models minus HadISDH. j, k, l) Difference (model minus observation) between HadGEM3-A and HadISDH for air temperature, specific humidity and relative humidity respectively. m, n, o) Difference (reanalysis minus observation) between ERA-Interim and HadISDH for air temperature, specific humidity and relative humidity respectively. All climatologies have been calculated over the 1975 to 2005 period using identical spatio-temporal coverage to HadISDH, except for ERA-Interim which starts in 1979.

The Southern Hemisphere is the region where the observations and models differ most in terms of the regional average time series (Sect. 4). However, climatologically, there is not a strong consistent difference between the historical forced CMIP5 models and HadISDH over this region. Some gridboxes appear to be generally warmer, moister and more saturated in the CMIP5 models (but not HadGEM3-A) than HadISDH, but other grid boxes are cooler, drier and less saturated. Areas of southern Africa, southern Australia and the west coast of South America appear to be cooler, contain more moisture, and be more saturated than HadISDH. Over Antarctica, the few gridboxes present are also cooler and more saturated, but not necessarily more moist. The middle of South America is generally warmer, drier and less saturated. Similarly for HadGEM3-A and ERA-Interim, while the majority of the observed Southern Hemisphere gridboxes are too highly saturated, the Southern Hemisphere does not stand out as a region of large climatological differences compared to HadISDH. As these features are very localised, it is not possible to say that these regions are biased compared to the observations.

HadGEM3-A is generally cooler than HadISDH overall, drier and less saturated over the tropics and more saturated over the mid-high latitudes, consistent with majority of the CMIP5 models. ERA-Interim shows better agreement with HadISDH overall, but has generally similar biases to HadGEM3-A, albeit to a lesser extent and with less spatial consistency. Dry biases relative to synoptic surface observations are also noted in Simmons et al. (2010).

Clearly, there are some notable differences between HadISDH and the other datasets (CMIP5 models and ERA-Interim) climatologically speaking, especially for relative humidity. Most importantly perhaps is the tendency for all models (CMIP5 and HadGEM3-A) to be too highly saturated in the mid-high latitudes and too arid in the low latitudes. Given these differences we expect differences in the spatial distribution of trends, which if large could impact large-scale average timeseries and trends, as well as $T - q$ and $T - rh$ relationships (Sect. 6).

## 5.2 Long-term trends

HadISDH spans from 1973-2015 inclusive, but the CMIP5 models mainly stop in 2012, with the few without *historicalExt* runs ending in 2005. We therefore calculate linear trends over 1975-2010 requiring 80% completeness. The trends are again calculated using the median of pairwise slopes method. For models which have no data post-2005 (IPSL-CM5A-LR and CSIRO-Mk-3-6-0 for *historical* humidity), these are still included in these calculations. The spatial and temporal coverage has been matched to that of HadISDH. We also calculate the number of historical CMIP5 model runs have trends greater than (more positive) or less than (more negative) those of HadISDH for the three variables, as well as whether the trend is in the same or opposite direction to HadISDH.

Despite the relatively poor coverage in the tropics, the relative humidity trends indicate increasing saturation in the tropics, and also the high latitudes. In the mid-latitudes, however, there are large areas where the relative humidity has declined. These regions extend further north in Europe than they do in North America and East Asia (Fig. 6c). In comparison, specific humidity shows an increase in moistness almost globally, with the largest increases over the tropics, though the Mediterranean region also stands out. Only a few grid boxes show decreasing trends, at the southern tips of the Southern Hemisphere land masses and also on the west coast of North America (Fig. 6b). The global temperature has increased in all but a handful of grid boxes, with

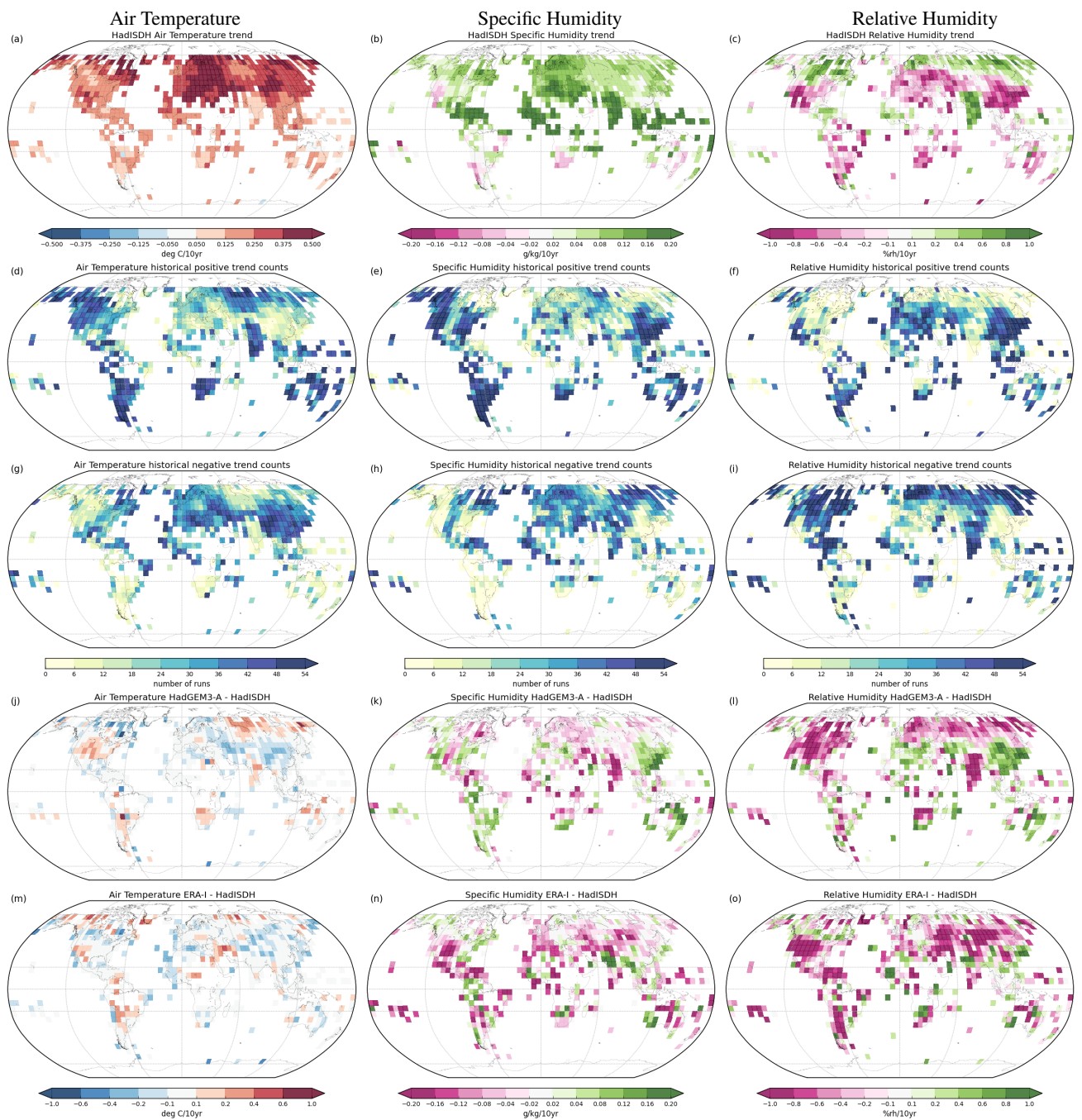

**Figure 6.** Differences in linear trends between models and observations at the gridbox scale. a, b, c) The HadISDH trends for comparison purposes (1975-2010 with 80% completeness). Frequency of historically forced CMIP5 models with d, e, f) positive trends (more warming/less cooling, more moistening/less drying, becoming more saturated/a lower rate of becoming more arid) or g, h, i) negative trends (opposite to above directions) relative to HadISDH for air temperature, specific humidity and relative humidity respectively. For a) to f) shading scales with the multi-model mean difference of models minus HadISDH. j, k, l) Difference (model minus observation) between HadGEM3-A and HadISDH for air temperature, specific humidity and relative humidity respectively. m, n, o) Difference (reanalysis minus observation) between ERA-Interim and HadISDH for air temperature, specific humidity and relative humidity respectively. All trends have been calculated using the median of pairwise slopes method, with identical spatio-temporal coverage to HadISDH.

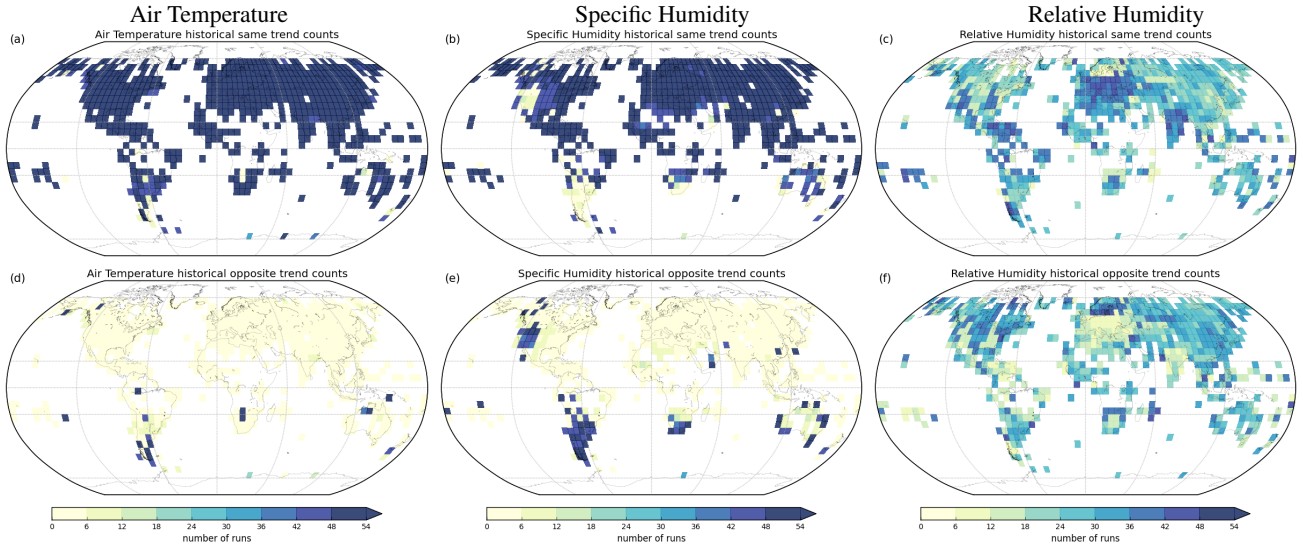

**Figure 7.** Frequency of historically forced CMIP5 models with trends in the a, b, c) same direction as HadISDH or d, e, f) opposite direction to HadISDH for air temperature, specific humidity and relative humidity respectively.

the strongest warming over eastern Europe and western Asia, and the Northern Hemisphere warming more than the Southern (Fig. 6*a*).

There is generally good agreement between the CMIP5 models and HadISDH for both the air temperature and the specific humidity trends. The strongest warming is in the northern high latitudes, and the strongest moistening is in the tropics. There
is a quasi-zonal pattern in the air temperature trend differences between the CMIP5 models and HadISDH, with stronger warming across the high latitudes, and weaker warming across the mid latitudes (Fig. 6*e, h*). The tropics show mixed signals, with many models showing stronger warming over India and south-east Asia, and weaker warming over tropical west Africa. The specific humidity in contrast shows stronger moistening over the Americas, east Asia, Australia and southern Africa, and weaker moistening over Eurasia, eastern North and Central America and northern Africa. As only a few boxes in both air
temperature and specific humidity have trends opposite to HadISDH (Fig. 7) this shows very good agreement in general in terms of the spatial patterns in direction of trends.

But many of the regions which have very strong moistening in HadISDH have weaker moistening in the CMIP5 models and also stronger warming in these areas. The few areas showing drying in HadISDH are also not replicated in the CMIP5 models, and are also seen in Fig. 7*b, e* as areas with opposite trends to HadISDH. However, HadGEM3-A shows some areas which
do exhibit drying in similar (though not exactly the same) locations as HadISDH (southwest USA, southern South America, southern tip of Africa, but not southern Australia; see Fig. 6*b* and Supplementary Information Fig. 31).

A number of CMIP5 models also show a few isolated grid boxes with a negative trend in the specific humidity (e.g. CSIRO-Mk3-6-0, HadGEM2-ES, Supplementary Information, Fig. 28). However these are most likely the result of artifacts of missing data in the observations which, when applied to the model values, results in poor temporal sampling in these very few gridboxes.

For relative humidity, there are spatially cohesive regions of both positive and negative trends in both the CMIP5 models and HadISDH, and weak trends towards less saturation are reasonably widespread in all models (compare Figs. 6*f, i* and 7*c, f*). More negative trends (decreased rate of saturation than HadISDH) over the USA, mid- and northern South America, southern Africa, northeast Europe/western Asia and India are common to most models (Fig. 6). Although India is becoming more saturated in the CMIP5 models (but not HadGEM3-a), it is doing so at a slower rate than HadISDH (see Fig. 7). More positive trends (increased rate of saturation than HadISDH) over southern Europe and eastern China are also common to most models. The drying regions in HadISDH take on a more zonal structure with the drying mostly across the mid-latitudes in both hemispheres.

Broadly, over the mid-latitudes, where HadISDH shows a trend towards less saturation (negative relative humidity), most CMIP5 models show more positive trends (Fig. 6*f, i*). Mostly this means that the model trends are still negative, but weaker than HadISDH, but in some cases the model trends are positive, especially over eastern China (Fig. 7*f*). Over the tropics and high latitudes, where HadISDH shows a trend towards increasing saturation (positive relative humidity), most CMIP5 models show more negative trends (Fig. 6*c, i*). This is a result of mostly weaker positive CMIP5 trends, or, especially over the high latitudes, negative trends. When comparing the direction of trends (Fig. 7*c, f*) this is also clear, with CMIP5 models being split between having trends in the same or opposite direction as HadISDH. The relatively high interannual variability of relative humidity in the CMIP5 models compared to the amplitude of the long-term trends contributes to this more mixed signal.

HadGEM3-A, as for temperature and specific humidity, shows similarities to the CMIP5 models, exhibiting some zonal banding of the differences. It does not show the trends of increasing saturation over the high latitudes observed in HadISDH. The prominent increasing relative humidity over India observed in HadISDH and a number of CMIP5 models is not present in HadGEM3-A.

In contrast the differences between HadISDH and ERA-Interim are more mixed, with few areas showing strong differences in the temperature trends. Moderate differences are apparent with more warming in northern Europe/western Asia, and cooling in western Europe/western Africa as well as an area in central Asia. For the humidity measures, the predominant signal is for a relative drying/less saturation compared to HadISDH, especially in the deep Tropics, western North America and western Europe. Note the changes in ingested SSTs in June 2001 and January 2002 in ERA-Interim that lead to a small downward shift in SSTs (see Section 2.4). This could be a contributing factor to the smaller positive land specific humidity trends and larger negative relative humidity trends. And in the regional average time series presented (Figs. 2 to 4), the largest differences between HadISDH and ERA-Interim are observed at that time across all variables.

## 5.3 Summary of spatial differences

The comparison of spatial patterns in the previous section focusses both on the background annual climatology and long-term linear trends. We note that we have not assessed any spatial correlations between the observations, *historical* models and reanalysis products.

For air temperature, the CMIP5 models are on the whole too cool over most of the globe, but parts of the mid latitudes (especially a band stretching eastwards from the Mediterranean Sea). Similarly, the CMIP5 models are too dry in the most part, but areas in the mid-latitudes appearing to be too moist. Conversely, the models are more saturated than the observations

in the high latitudes, but too arid in the tropical regions. Across all three variables, there is a suggestion of a zonal pattern - with mid latitudes showing a different signal to the rest of the globe in the temperature and specific humidity, and a stricking contrast between the tropics and high latitudes in the relative humidity. However the Southern Hemisphere, which stood out in the temporal analysis, does not show strong, consistent differences in any of the three variables. HadGEM3-A is on the whole a little cooler and drier than HadISDH, and less saturated in the tropics, but more saturated over mid-high latitudes. ERA-Interim shows a similar pattern, but with lower magnitudes.

Comparing long-term trends to the models shows general good agreement between the CMIP5 models and HadISD for air temperature and specific humidity. The direction of the model trends are well aligned to the observed trends for the majority of the globe for these two variables, (Fig 7) though there are regional differences on the magnitude of the trends, and a quasi-zonal pattern, especially in the eastern hemisphere. Many of the regions showing strong moistening in the observations show weaker moistening in the CMIP5 models, combined with stronger warming. For relative humidity, there is not even agreement on the direction of the trends. Again, there is an apparent zonal signal, with models having negative trend differences in high latitudes and India, but positive ones in the eastern hemisphere mid-latitudes. In both HadGEM3-A and ERA-Interim the largest differences are for the relative humidity trends, with the smallest in air temperature. Both these products show shallower trends in relative humidity in most regions.

Away from the high latitudes, the relative humidity trends in the historical CMIP5 models look similar to the RCP8.5 multi-model ensemble mean difference between 2071-2100 and 1971-2000. Relative humidity is projected to become less saturated over most of the land mass apart from regions around India, parts of tropical Africa and the southern Arabian peninsula. The more zonal pattern in the observations includes increasing saturation levels over the Caribbean, western Africa and the high northern latitudes are not present in the future projected changes (Collins et al., 2013, Fig. 12.21, Sherwood and Fu, 2014, Fig. S1). Thus, despite some similarities, the present observed drying appears to be different to the long-term projected trends of drying. This suggests the importance of some decadal scale variability or driving mechanisms that are not well represented in GCMs such as land use change.

There is no evidence that climatologically moister/more saturated regions have stronger moistening trends/less strong or decreasing saturation trends, so we cannot conclude that the models are less water limited than the observations from this analysis. Just as with the time series, agreement between the models and observations is better for temperature and specific humidity than for relative humidity. There is arguably better agreement in terms of spatial patterns of drying than the long-term large-scale average time series. This shows how linear trends mask the significant temporal differences that can be seen in the time series. Despite the same modes of variability and SSTs, the spatial pattern of trends in HadGEM3-A do not match the observations very well. The spatial differences to projected future trends in relative humidity suggest that for those regions (Caribbean, high latitudes) a number of possible issues are present: there could be larger observational errors, processes not captured well in the models, more transitional processes associated with the land-surface or even modes of variability that could not be expected to persist to centennial time scales nor be captured by climate models.

## 6 Comparison of relationships in temperature and humidity between observations and models

Following our investigations into the temporal, spatial and spatio-temporal behaviour of the modelled and observed temperature and humidity variables, we turn to an inter-variable analysis, assessing correlations between the variables themselves. The Clausius-Clapeyron relationship indicates that for a larger increase in temperature, there will be a proportionally larger increase in specific humidity, as long as water availability is not limiting. Furthermore, we expect a larger increase in specific humidity for a $1°C$ rise in temperature in the warm tropics (or a warmer background climatology) than over the cooler high northern latitudes (or a cooler background climatology). We can explore this explicitly by looking at the relationship between temperature and specific humidity, and between temperature and relative humidity across the different models, forcing scenarios and regions.

Dai (2006) showed that the relative and specific humidity were over-correlated with the temperature fields in a study of CMIP3 models. By using the annual temperature and humidity anomalies from Fig. 2 to 4, we can compare the behaviour of the CMIP5 models against the observations. As the agreement between the models and observations differs depending on the region, we will also explore the degree to which temperature-humidity relationships differ from region to region. Figs. 8 and 10 show the temperature–specific humidity relationship and temperature–relative humidity relationship respectively for the globe and Southern Hemisphere and a selection of CMIP5 models. Further plots for all models and regions can be found in the Supplementary Material (Figs. 19-23 and 33-37). The HadGEM3-A versus HadISDH and ERA-Interim relationships are shown in Fig. 9, also for the globe and Southern Hemisphere.

We note that in this section, all model realisations are shown (across all three experiments) compared to the single realisations of the observations and reanalyses. This is a necessary step for using the ensemble means would smooth out some of the inter-annual variability and reduce the power of this asssessment.

### 6.1 Temperature-specific humidity relationship

Both HadISDH and ERA-Interim exhibit positive temperature-specific humidity ($T - q$) relationships for all regions. They behave reasonably linearly. The steepest observed $T - q$ relationship slopes are in the tropics. This makes sense, given that the tropics (20°S to 20°N) is the warmest region – a warming trend will drive larger moistening trends there. The smallest observed $T - q$ relationship slopes (close to zero) are in the Southern Hemisphere. Correlation is also lowest in the Southern Hemisphere to the extent that there is no real relationship between temperature and specific humidity there, the observed regions of the Southern Hemisphere appear very water limited. ERA-Interim has smaller slopes and weaker correlations compared to HadISDH but they are broadly similar. The strongest correlations occur in the tropics for HadISDH and in the Northern Hemisphere for ERA-Interim (see Supplementary Material, Figs 34 and 35). The location of individual years along with their relative humidity anomaly is shown in the Supplementary Material, Fig. 37.

Overall, the majority of *historical* forced CMIP5 models exhibit slightly steeper (more positive) temperature-specific humidity relationship slopes, with stronger correlations than HadISDH or ERA-Interim (Fig. 8). This supports the concept of $T$ and $q$ being more closely correlated in the models than the observations. However, this is partly to be expected given the stronger

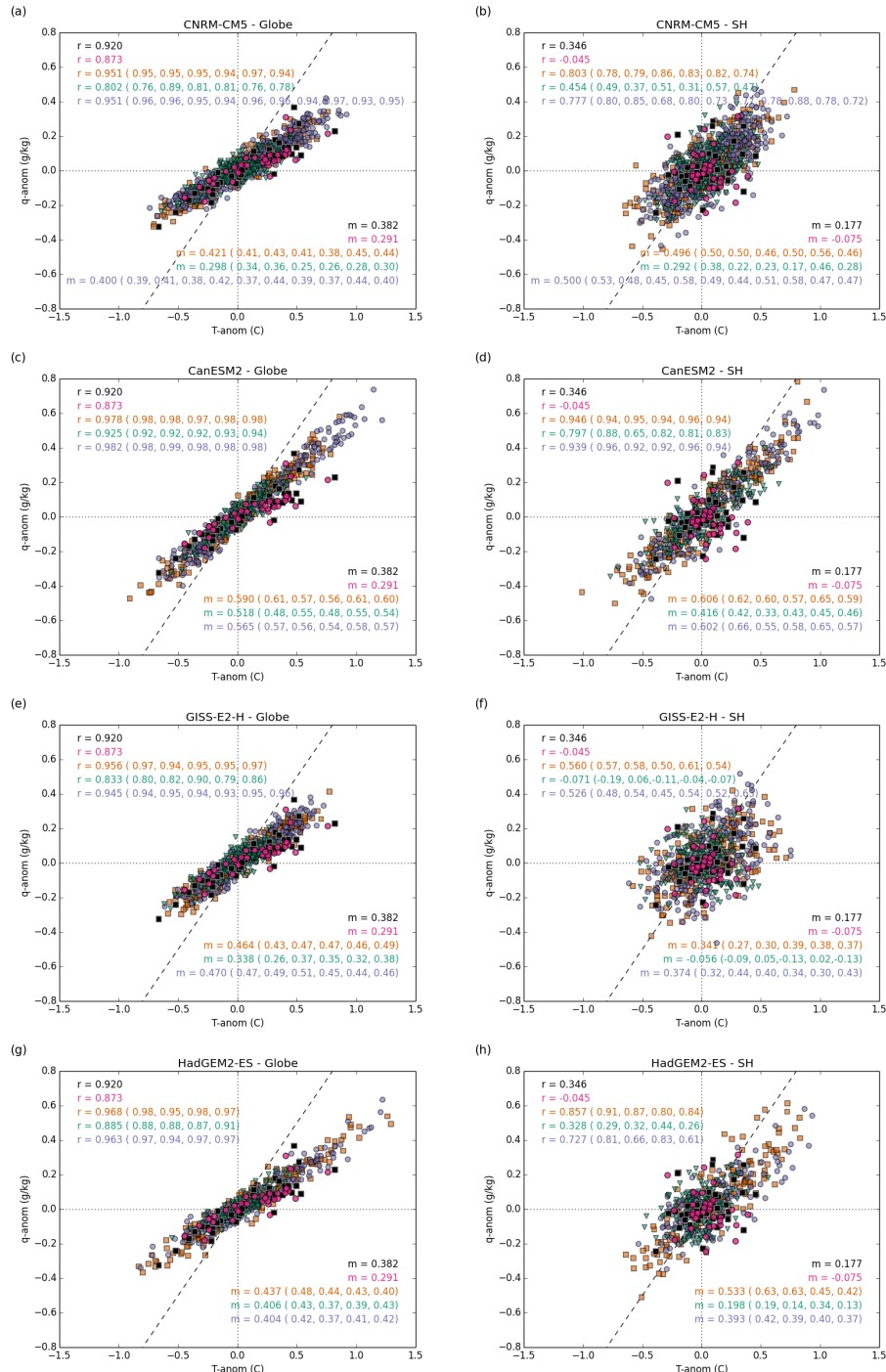

**Figure 8.** [Panel sizes reduced to fit Discussion Paper] Relationships between temperature and specific humidity for the globe (a, c, e, g) and southern hemisphere (b, d, f, h) for a selection of CMIP5 models compared to HadISDH and ERA-Interim. Correlations (r) are shown in the top left hand corner. The gradient of the line of best fit is given in the bottom right hand corner. HadISDH is shown in black. ERA-Interim is shown in magenta. *Historical*, *historicalNAT* and *historicalGHG* are shown in purple, green and orange respectively. The ensemble mean value is given for each model with individual member values in parentheses.

warming in the *historical* CMIP5 models. Clearly, the slight cool bias in the *historical* CMIP5 climatological temperature is not large enough to significantly change the $T - q$ relationship slope.

This tendency for larger slopes and stronger $T - q$ relationships in the *historical* CMIP5 models is true for all regions, especially the Southern Hemisphere (Fig. 8). Notably, CSIRO-Mk3-6-0 is unique in its similarity to the observations in the Southern Hemisphere. HadGEM2-ES and CSIRO-Mk3-6-0 have slopes closest to HadISDH, and these slopes are slightly smaller than those in HadISDH in the tropics. Like the observations, the CMIP5 models consistently show the largest slopes in the tropics (apart from NorESM1-M which places the Southern Hemisphere just above the tropics but from few ensemble members) and the majority of *historical* CMIP5 models also have the highest correlations there as well. CSIRO-Mk3-6-0, CanESM2 and HadGEM2-ES have the strongest correlations in the Northern Hemisphere or globe along with ERA-Interim. However, unlike the observations all *historical* CMIP5 models apart from CSIRO-Mk3-6-0 have a larger $T - q$ relationship slope in the Southern Hemisphere than for the globe and Northern Hemisphere, though the correlation coefficients are in some cases relatively low ($\sim 0.3$). Overall, the larger slopes and higher correlations in the CMIP5 models suggests that water availability could be less of a limiting factor in the models compared to the observations, especially in the Southern Hemisphere (see also Section 4). This is supported to some extent by the prevalence of overly moist and saturated *historical* CMIP5 model gridboxes in parts of the Southern Hemisphere as discussed in section 5.1.

Interestingly, most *historicalNAT* CMIP5 models appear more consistent with HadISDH and ERA-Interim over the Southern Hemisphere in terms of small slopes and low correlations. The *historicalNAT* CMIP5 model Southern Hemisphere correlations are notably lower than for other regions, but slopes are not consistently smaller. The difference between and within the *historicalNAT* CMIP5 models suggests that natural variability does contribute to differences in the temperature-specific humidity relationship. However, the consistency between models, which collectively explore a wide range of natural variability at any one point in time, suggests that Southern Hemisphere differences are not strongly driven by natural variability.

It appears that the human forcing component in the models is contributing to humidity changes in the Southern Hemisphere that are not consistent with the observations. We have established that the models are overall biased cool and dry (in absolute moisture terms) relative to the observations, suggesting that climatological biases are not driving differences in the global average.

In all cases, the *historical* forced CMIP5 models have larger warming trends than the observations which could be a contributing factor, especially as the trend differences tend to be greatest in the Southern Hemisphere. However, for some models and regions the $T - q$ relationship slope is largest in the *historicalNAT* forced ensemble mean relative to the *historical* forced runs, even though *historicalNAT* warming trends are much smaller than *historical* warming trends. This is the case for HadGEM2-ES and bcc-csm1-1 over the globe and Northern Hemisphere, and for CSIRO-Mk3-6-0 over the Northern Hemisphere, although there is an overlap between the range of slopes of the $T - q$ relationship from the individual runs between *historical* and *historicalNAT*. This suggests that for these models at least it is not just the stronger warming trend in the CMIP5 models that is driving the stronger relationship and that natural variability may play a role in determining how specific humidity changes over time.

The atmosphere only HadGEM3-A model for the most part has the same patterns of natural variability as the observations. Trends and climatology are fractionally more similar to HadISDH than for the *historical* CMIP5 models. As expected, the temperature-specific humidity relationship slopes and correlations are substantially more similar to HadISDH and ERA-Interim (Fig. 9). The ensemble mean $T-q$ relationship slope for the globe and Northern Hemisphere is slightly smaller than for the observations, as is the spread of the individual runs. For the tropics the ensemble mean $T-q$ relationship slope is slightly larger than for HadISDH and ERA-Interim. Again the Southern Hemisphere is the region where differences are largest. Although the correlations are similar, and the ensemble range of slopes and correlations encompass the HadISDH observed values, the ensemble mean slope is slightly more positive in HadGEM3-A. It is actually more positive than for the Northern Hemisphere, which is not the case in HadISDH or ERA-Interim, but has a lower correlation indicating that this has lower robustness. Overall, this good agreement suggests that the model physics is reasonable outside of the Southern Hemisphere – when background climatology, variability and trends are similar, the models behave very similarly to the observations in terms of large-scale physical relationships.

Overall, the $T-q$ relationships are reasonably similar to the observations, except for the Southern Hemisphere. Generally the correlations are a little stronger in the CMIP5 models, with steeper slopes. However in the Southern Hemisphere these differences are larger in CMIP5 and even for HadGEM3-A. Apart from observational error being a possible cause, these differences also suggest some issues with the model physics in the Southern Hemisphere

## 6.2 Temperature-relative humidity relationship

In a region that is not water limited, the expectation is that relative humidity should not change with temperature, the slope and correlation should be very close to zero. Given that water availability is limited over land to some extent, and additionally that the land has been warming faster than the ocean, a negative $T-rh$ relationship might be expected. The $T-rh$ relationship slopes are negative for all regions for both HadISDH and ERA-Interim and largest (most negative) by a long margin in the Southern Hemisphere (Fig. 10). This is to be expected given the lack of relationship between temperature and specific humidity for this region, again suggesting that it is strongly water limited. ERA-Interim consistently has more negative slopes and stronger (more negative) correlations than HadISDH. The strongest correlation for HadISDH is in the Southern Hemisphere but still quite weak at -0.550. For ERA-Interim, all correlations are more negative than -0.5 and strongest (most negative) for the globe.

The scatter pattern of both HadISDH and ERA-Interim for the globe and Northern Hemisphere is distinctly different from anything shown in the models (including HadGEM3-A), and very clearly non-linear (Fig. 10, most clearly seen against models with few ensemble members). For both HadISDH and ERA-Interim there appear to be two populations. For all times where temperature anomalies are lower than $\sim 0.2^\circ C$ there appears to be a neutral or small positive $T-rh$ relationship. For all temperature anomalies greater than $0.2^\circ C$ there is either a very large negative or a neutral $T-rh$ relationship. Either way, this second cluster appears to behave differently to the first. Also, this behaviour is not apparent in the Tropics or Southern Hemisphere. In Supplementary Material Fig. 23, the $T-rh$ relationship is shown along with the years and the specific humidity

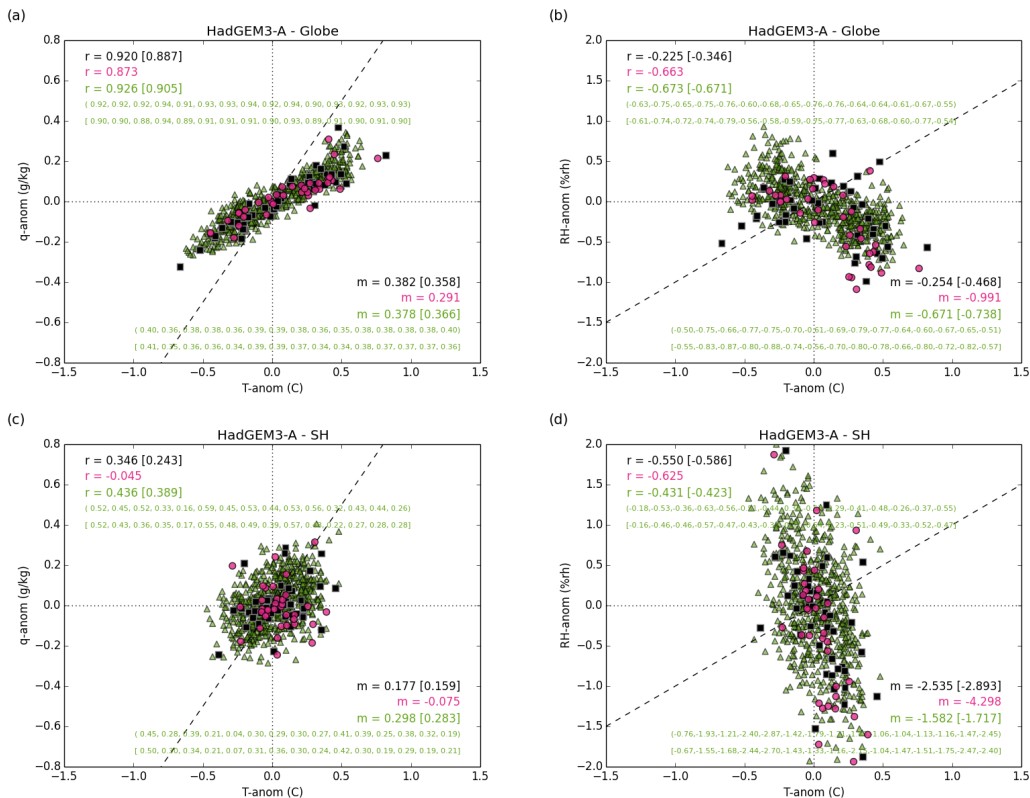

**Figure 9.** Relationships in Global (a, b) and Southern Hemisphere (c, d) average temperature and specific humidity (a, c) and temperature and relative humidity (b, d) for HadGEM3-A compared to HadISDH and ERA-Interim. Correlations (r) are shown in the top left hand corner. The gradient of the line of best fit is given in the bottom right hand corner. HadISDH is shown in black. ERA-Interim is shown in magenta. The ensemble mean value is given for HadGEM3-A with individual member values in parentheses. Values in square brackets are calculated using data matched to the coverage of ERA-Interim, for both HadISDH and HadGEM3-A.

anomalies. The positive specific humidity anomaly years (mainly the most recent years) seem to have a different relationship to the negative anomaly years (mainly the earlier years) for the Northern Hemisphere, and to some extent in the Tropics.

In general, the $T-rh$ correlations for all models, all regions and the observed estimates are low (Fig. 10). This shows that the relationship between temperature and relative humidity is much weaker than for temperature and specific humidity, consistent across all models and observations, and that the slopes should be interpreted with caution. However, none of the CMIP5 models exhibit anything like the scatter patterns shown in HadISDH or ERA-Interim. Nevertheless, some interesting common features are apparent. Overall, the majority of *historical* forced CMIP5 model ensemble members exhibit negative temperature-relative humidity relationships. However, here is far greater variation in the $T-rh$ relationships across the *historical* CMIP5 models compared to the $T-q$ relationships, with slopes and correlations both steeper/stronger and shallower/weaker than HadISDH. Interestingly, the CMIP5 models exhibit consistently shallower slopes than ERA-Interim. In the Southern Hemisphere, the

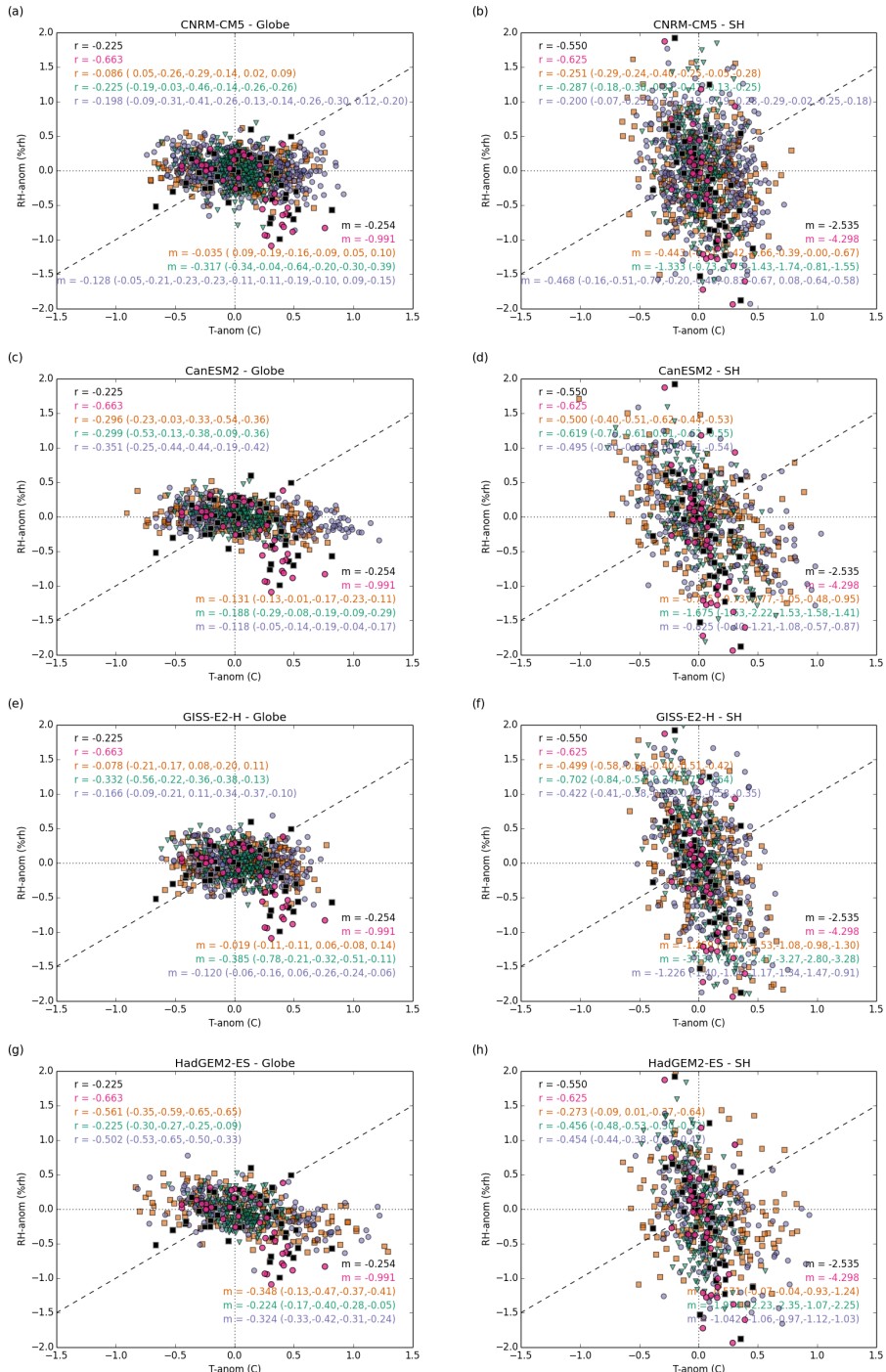

**Figure 10.** [Panel sizes reduced to fit Discussion paper] Relationships between temperature and specific humidity for the globe (a, c, e, g) and southern hemisphere (b, d, f, h) for a selection of CMIP5 models compared to HadISDH and ERA-Interim. Correlations (r) are shown in the top left hand corner. The gradient of the line of best fit is given in the bottom right hand corner. HadISDH is shown in black. ERA-Interim is shown in magenta. *Historical*, *historicalNAT* and *historicalGHG* are shown in purple, green and orange respectively. The ensemble mean value is given for each model with individual member values in parentheses.

slopes for the CMIP5 models are on the whole more negative than they are in for the globe and for other regions, as in HadISDH and ERA-Interim (except for GISS-E2-H and IPSL-CM5A-LR). The slopes are mostly more negative than HadISDH for the Northern Hemisphere but shallower than ERA-Interim. The models have shallower negative slopes than HadISDH for the Southern Hemisphere and very mixed for the Tropics, with weaker correlations.

HadGEM3-A appears to be closer to ERA-Interim for the Globe and Northern Hemisphere than the other models or the observations. The CMIP5 models are closer to the observations for the Globe and Northern Hemisphere than HadGEM3A (Fig. 9). HadGEM3-A is within the model spread for the tropics and Southern Hemisphere, along with HadISDH. ERA-Interim on the other hand, is with the model spread for the tropics but not for the Southern Hemisphere.

## 6.3   Summary of $T - q$ and $T - rh$ relationships

As outlined previously, the specific humidity is closely linked to the surface temperature (assuming no water limitation) and so a relatively tight correlation is expected, and also seen all regions apart from the Southern Hemisphere. However, as relative humidity is a more sensitive variable and also no trend is expected under the same assumptions, then a higher apparent scatter would be expected. Combined with the necessity of showing each model and experiment realisation compared to the single realisations of HadISDH and ERA-Interim, the qualitative differences in the appearance of the $T - q$ and $T - rh$ plots follow.

We have discussed previously whether the models could have different levels of water limitation compared to the observations when analysing the temporal behaviour over large regions (Sect. 4), but that overall differences are not clearly evident from the distribution of differences in trends and climatologies (Sect. 5). The presence of a negative $T - rh$ relationship shows that the models are water limited to some extent. For the Northern Hemisphere and Globe, the generally stronger correlations and steeper negative slopes in the *historical* CMIP5 models suggest that the models are more water limited than the observa-

tions in these regions. Conversely, for the Southern Hemisphere, and to some extent in the Tropics, the weaker correlations and shallower negative slopes in the *historical* CMIP5 models suggest that the models are less water limited than the observations. However, this interpretation does not take into account the complex interdecadal behaviour of the observation based estimate, especially for the Northern Hemisphere and Globe, which is very different to the more linear relationship that is consistently shown across the models. Furthermore, climatological differences point to *historical* CMIP5 models generally being drier and

less saturated across the tropics, which might suggest that they are more water limited than the observed estimates. The higher latitudes appear to be more saturated than the observations, but only polewards of $40°$ N/S. The hemispheric averages included data down to $20°$ N/S which would dampen any expected signal. For all other regions, the range of slopes from individual *historicalNAT* CMIP5 model runs encompasses the *historical* CMIP5 model ensemble mean. This suggests that natural variability also plays a strong part in the $T - rh$ relationship over these large scales.

We would expect better agreement between the observed estimates and HadGEM3-A than for the CMIP5 models. However, this does not appear to be the case. For the globe and Northern Hemisphere the slopes and correlations in HadGEM3-A are much more negative than those in HadISDH, and many of the *historical* CMIP5 models, although they are closer to those in ERA-Interim (Fig. 9).

For the tropics, where there is large variability across the CMIP5 models, and the Southern Hemisphere, *historical* HadGEM3-A and HadISDH are both within the coupled model spread. However, like the CMIP5 models, HadGEM3-A shows shallower negative slopes in the Southern Hemisphere suggesting that it is less water limited than the observed estimates there.

As in the other analyses, the agreement between models and observations is generally poorer where relative humidity is concerned compared to specific humidity. There is consistency across the CMIP5 models in terms of their negative $T - rh$ relationships, the largest negative relationship occurring in the Southern Hemisphere. In this region, and in the tropics, there is a larger negative relationship in the *historicalNAT* ensemble members compared to *historical*. The range of slopes from the *historicalNAT* runs is mostly wider than that of the *historical* runs, encompasses the ensemble mean *historical* slope and overlaps with the *historical* slope range in most cases for globe and Northern Hemisphere. This is not the case for the tropics or Southern Hemisphere though. Hence, natural variability appears to be a larger contributing factor than greenhouse gases to the variation observed in the global and Northern Hemisphere averages. The smaller negative slopes in the Southern Hemisphere and also the tropics for historical CMIP5 models relative to *historicalNAT* suggest that in the models, human activity may be driving a weakening of the $T - rh$ relationship, making the region less water limited.

While the agreement between the models themselves and between the models and observations is good for $T - q$ (outside of the Southern Hemisphere), there is much poorer agreement both between the models themselves and between the models and observations for $T - rh$. In general the $T - q$ relationship is strongly positive, whereas the $T - rh$ relationship is weakly negative. The Southern Hemisphere appears unique in that the $T - q$ relationship is generally weakly positive whereas the $T - rh$ relationship is generally strongly negative, for both models and observations. The CMIP5 models consistently have a stronger $T - q$ correlation that is more steeply positive than HadISDH, ERA-Interim and HadGEM3-A, even in the Southern Hemisphere. Also, reasonable agreement between ERA-Interim and HadISDH suggests some robustness in the observed features. This over-correlation as discussed in Dai (2006) could be due to missing processes in the models or errors in the observations or models. As this analysis uses spatially and temporally matched data it is not the result of coverage issues. The over-correlation may be the result of the models sampling slightly different climatologies for the regions discussed above. If the coverage is sparse and the few areas sampled are climatologically different, then different relationships may be expected.

For $T - rh$, the CMIP5 models (*historical* and *historicalNAT*) have much wider spread that encompasses HadISDH in terms of correlation strength and slope steepness, but is consistently weaker/shallower in the Southern Hemisphere. This suggests a high sensitivity of relative humidity to model parameterisations or natural variability. Generally over all regions, the *historicalNAT* $T - q$ correlations/slopes from CMIP5 are weaker/shallower than the *historical* ones. This suggests some strengthening of the $T - q$ relationship driven by anthropogenic climate change. In the tropics and especially the Southern Hemisphere, the *historicalNAT* $T - rh$ have stronger correlations/steeper slopes than *historical* experiments, which is consistent with the weaker/shallower *historicalNAT* $T - q$ relationship. This weakening of the $T - rh$ relationship under anthropogenic climate change, while again largely driven by the presence of a strong trend in temperature, suggests less water limited conditions under anthropogenic climate change.

## 7 Discussion

In the preceeding sections we have presented assessments of contrasts and similarities between the observations and models using temporal, spatial and spatio-temporal information for the temperature and humidity variables, as well as the level of correlation between the variables themselves. We now draw these strands together to pull out common threads in these different analyses.

The long-term trends of regional scale averages (Sect. 4) show some general agreement between the CMIP5 models and HadISDH for temperature and specific humidity. The similarity in long term trends over the entire period indicate the increasing concentrations of greenhouse gases are at least part of the story for these two variables. However differences in the most recent period are not fully explained by these increasing concentrations. In general, there is better agreement between the CMIP5 models and the observations, and between the models themselves, for temperature than for the humidity measures. Similarly the agreement is better for specific humidity than for relative humidity.

The Southern Hemisphere has more models where the *historical* experiments do not match the observations, whereas the experiments without greenhouse gas forcings (*historicalNat*) do. This may be linked to the cooling noted in the Southern Ocean in both *historicalNat* experiments and the observations (Jones et al., 2013), and may be driven by the Southern Annular Mode. If the surface temperature behaviour differs, then this is likely to feed through into the specific humidity. A cooler ocean results in less moisture being advected over land, and a lower specific humidity. However, there is no clear difference in relative humidity, maybe because of the relatively small land fraction in the Southern Hemisphere (and data coverage effects) as well as a slower warming rate in this region.

The changes observed in behaviour of the relative and specific humidities in the recent decade or so are coincident with an apparent reduction in the rate of global (combined land-air and sea) temperature rise (Cohen et al., 2012; Hartmann et al., 2013; Kosaka and Xie, 2013). During this period, also referred to as a "hiatus" or slow-down, the land surface air temperature continued to show a warming trend, at a faster rate than that over the ocean. However the combined rate of temperature rise was lower than in the previous decades. Some of the causes postulated were increased ocean heat uptake (Katsman and van Oldenborgh, 2011; Meehl et al., 2011, 2013), solar effects (Hansen et al., 2011), changes in atmospheric water vapour (Solomon et al., 2010) or aerosols (Solomon et al., 2011) and increased wind-driven circulation in the Pacific (England et al., 2014). Recent analysis of the NOAA surface temperature product Karl et al. (2015); Hausfather et al. (2017) shows very little difference in the linear trends over time, but the magnitude does depend on the periods chosen. By comparing several datasets, Simmons et al. (2016) show that those which provide more values in the polar regions indicate that 2016 is warmer than 2015, but those which do not suggest these two years were similarly warm. Their estimates of trends over this recent period (1998-2012) are higher than the central estimate from the IPCC AR5 (Hartmann et al., 2013), suggesting less of a "hiatus" over this period than earlier analyses. Other assessments have reached similar conclusions (Rahmstorf et al., 2017; Medhaug et al., 2017)

In more recent years, global average temperatures have reached record values, warmer than the El Niño year of 1998 (Sanchez-Lugo et al., 2016), and then 2016 being the hottest year in the instrumental record WMO (2017). Similar behaviours

(short term increases/decreases in the warming rate superimposed on a longer-term behaviour) have been observed in the past. In general, however, it is expected that short period trends vary around trends measured over a longer period, with short period trends lying within the natural climate variability (see e.g. Koutsoyiannis and Montanari, 2007).

In a warmer climate, if the amount of moisture available over land reduces, then there is greater sensible heating as a result. Therefore, although specific humidity rises with increasing temperatures, it rises less over land than oceans (Sherwood and Fu, 2014). Moist enthaply characterises the energy content of a parcel of air from both temperature and humidity. Berg et al. (2016) show that the increase in the combined moisture and temperature (moist enthalpy) is constrained by the characteristics of the ocean in CMIP5 experiments, but not by soil moisture. However, aridity depends on both land surface processes (associated with decreasing soil moisture and leading to larger increases in sensible heating/temperature), and increased $CO_2$ fertilisation (leading to decreases in evapotranspiration through enhanced stomatal efficiency driven by increases in $CO_2$ concentrations). These two land-surface processes contribute strongly to temperature, specific and relative humidity changes at least on a local level and can amplify the aridity response. Without these land-surface processes, the modelled land-ocean temperature contrast results in declining relative humidity, but to a lesser extent (Berg et al., 2016). Hence, the degree to which climate models include soil moisture and evapotranspiration responses to $CO_2$ will affect the degree to which they can replicate observed rates of RH decline (Berg et al., 2016). Another explanation for part of the extra warming observed over land is from stomatal resistance responding to increased $CO_2$ by reducing evapotranspiration. The evaporative cooling of the surface (latent heating) is reduced, resulting in more energy available for sensible heating (Dong et al., 2009).

By using an atmosphere-only model in this study, an important aspect of the "hiatus" is prescribed within the experiment; the SSTs. When using HadGEM3-A, there is better agreement to HadISDH compared to the CMIP5 models for temperature and specific humidity over the latter half of the study period, corresponding to the "hiatus". However there is limited improvement for relative humidity, as HadGEM3-A shows a gradual decrease in relative humidity over the entire study period. The strong decrease in observed relative humidity in the most recent period is also not captured by this model, despite the improved agreement for temperature and specific humidity over the same time period. This indicates that relative humidity is a more sensitive measure than temperature or specific humidity alone, which follows as relative humidity compounds changes in both these variables. It is also likely that relative humidity is more affected by land-surface processes that only indirectly affect temperature and specific humidity. However, the improvement when using the atmosphere only model (albeit slight for relative humidity) over the CMIP5 ensemble indicates that the "hiatus" has a role to play in explaining the observed behaviour of land-surface humidity.

Furthermore, the prescribed SSTs will also result in an improved representation of some atmosphere-ocean circulation patterns, e.g. ENSO. As large-scale atmospheric circulation patterns play a role in the drying or moistening of a region, by capturing their behaviour and phase via the prescribed SSTs, there should be an improvement in the short timescale variability over the CMIP5 ensemble. This is indeed the case (Figs. 2 and 3), as both the long timescale and short timescale match to the observations is improved when using HadGEM3-A over HadGEM2-ES.

The origin of the atmospheric water vapour over the land surface is predominantly the oceans. If the ocean surface has experienced slower warming in recent years, the rate of increase in water evaporated and then held as a vapour in the air will

also have slowed. Hence, the rate of increase in marine specific humidity would be slower. As a major source of water vapour over land, this in turn will have affected the rate of increase in specific humidity over land. If the air over the land surface has warmed at a faster rate than over the oceans, the slower rate of increase in water vapour available for advection over land could result in an even slower rate of increase in specific humidity over land. The relative humidity over land would decrease

accordingly. This is because the amount of water vapour required for saturation will have increased with rising temperature but disproportionately to the actual increase in water vapour (Joshi et al., 2008; O'Gorman and Muller, 2010; Simmons et al., 2010; Sherwood and Fu, 2014; Chadwick et al., 2016). This is broadly what has been observed over the period from the end of the twentieth century. Changes in the large-scale circulation patterns would also affect the moisture availability over the land surface (Joshi et al., 2008). Changes in land cover and land use could have local effects, and this picture is further complicated

by the continuing rise in the marine specific humidity, which is not rising fast enough to provide enough moisture in the air advected over land to keep relative humidity constant (Berry and Kent, 2009, 2011; Willett et al., 2016). This analysis, however, is predominantly based on the Northern Hemisphere observations, so there is some uncertainty (Willett et al., 2016).

We have looked at climatologies to explore the degree to which the models are more or less water limited than the observations (Sect. 5.1). The CMIP5 models, and also in HadGEM3-A and ERA-Interim to some extent, are cooler everywhere,

drier/less saturated in the tropics to extra-tropics, and have comparable moisture levels but are more saturated in the high latitudes than the observations. We cannot conclude that the CMIP5 estimates are less water limited that the observations, but there are regional climatological differences which can be considerable. This is especially true for relative humidity where the anomalies can be greater than $\pm 10\%$rh.

As a result of these differences in the climatologies, we should expect differences in the spatial distribution of trends, which if

large could contribute to features in large scale average time series and trends and $T - q$ and $T - rh$ relationships. Interestingly, the Southern Hemisphere does not stand out despite the differences in the regional time series and trends discussed above, which may be due to the small land fraction polewards of 20°S.

For temperature and specific humidity the spatial pattern of trends are very similar (Sect. 5.2), although the small drying regions observed in the specific humidity are not represented in the CMIP5 members. HadGEM3-A does show some regions of

drying, but these are not identical (in location or size) to those observed. The largest differences occur in relative humidity, and these are considerable. The CMIP5 models show a lesser degree of moistening over the Tropics (and none in the Caribbean) and very little moistening over the high latitudes. The observed mid-latitude drying is present, but to a lesser degree than the observations. In general, HadGEM3-A spatial patterns are more similar to the CMIP5 models and future projected trends than to HadISDH.

This suggests that the drying identified in ERA-Interim and HadISDH could be due to processes not well represented in the models (e.g. land use changes, evapotranspiration response to $CO_2$, soil moisture changes), transitional features not associated with long-term climate change nor well represented in the models, or errors from the models or observations.

The observed trends of relative humidity appear to be arranged into zonal bands, with northern high latitudes becoming more saturated, mid-latitudes less, and parts of the tropics (Central America, West Africa, India) again becoming more saturated.

Therefore regions where CMIP5 modelled relative humidity does not increase as much as in the observations tend to have

experienced stronger warming. They also on the whole show less strong moistening relative to the observations (e.g. India and the high northern latitudes), but not in all cases (e.g. West Africa).

Both the initial investigations into the representation of humidity variables within the CMIP5 models compared to HadISDH along with the analysis of Dai (2006) led us to look at the $T-rh$ and $T-q$ relationships as there are suggestions that temperature and specific humidity are over correlated in the models. Our investigation (Sect. 6) shows that while agreement between the models themselves, and the models with HadISDH is good for $T-q$ (outside of the Southern Hemisphere), there is much poorer agreement in both cases and all regions for $T-rh$. In general the $T-q$ relationship is strongly positive whereas $T-rh$ is weakly negative. The Southern Hemisphere appears unique in that the $T-q$ relationship is generally only weakly positive where as the $T-rh$ is strongly negative. Also, the CMIP5 models consistently have a stronger $T-q$ correlation that is more steeply positive than HadISDH, ERA-Interim and HadGEM3-A, even in the Southern Hemisphere. This over-correlation may be the result of missing processes in the models or from errors in the models or observations.

Clearly, the CMIP5 models differ sufficiently to the observations to make any formal detection and attribution assessment of the humidity measure invalid. These differences arise from spatially and temporally matched fields of identical grid box size and climatology period, and manifest in the large scale average time series of specific and especially relative humidity, the climatological differences of both humidity measures, the spatial pattern in the relative humidity trends, the over correlation of $T-q$ and generally poor agreement between the models and between models and observations in $T-rh$. Furthermore, the lack of agreement in the Southern Hemisphere for $T-q$ and $T-rh$ is striking.

This assessment of the observed and modelled, both coupled and atmosphere-only, humidity behaviour opens a number of avenues for further investigation to determine the reasons behind the apparent mis-match in the recent past. As this is a land-only dataset, a similar assessment in the marine domain would be valuable, and a marine dataset is in development (Willett et al, in prep). Long model control experiments could be analysed to find periods of high contrast between the land and ocean heating, and what the behaviour of humidity is during that time. To further assess whether land-surface processes are involved, land-surface modelling experiments would build on the information that including the HadGEM3-A atmosphere-only model has had in this work, although a wider selection of AMIP models may also be useful in this regard. The zonal aspect of some of the humidity changes may be related to large scale changes in circulation patterns. And changes in humidity could be compared to changes in precipitation patters, both observationally and in models.

## 8 Summary

We have used the latest observational humidity dataset, HadISDH, to compare with the latest generation of climate models. We selected those models present in the CMIP5 archive which have suitable experiment runs (*historical, historicalNat, historical-GHG*) and which cover the most recent period. Global and regional average time series from these and also the ERA-Interim reanalysis were compared, as well as the climatologies, the spatial pattern of trends and $T-q$ and $T-rh$ relationships.

We have shown that while there is very broad agreement in large scale long-term changes, there are significant interdecadal, regional scale and physical relationship differences sufficient to mean that future modelled changes in specific humidity and

relative humidity over land are uncertain. Note that the observed relative humidity trends are spatially a little different to future RCP8.5 trends, particularly over the Caribbean and high latitudes. The main driver of the differences in recent trends in surface humidity is likely the differences in the observed and modelled changes in SST along with modes of variability. We used an atmosphere-only model to ensure similar trends in SST and modes of variability, which showed that there is better agreement, but still not sufficiently so for humidity in the one model we assessed. Although it showed generally declining relative humidity, the inter-decadal pattern of the observations was not well replicated, and neither was the spatial pattern. In particular, land related processes such as land surface type and change, soil moisture and evapotranspiration response to increasing $CO_2$ are possible contributors that help explain the differences that exist even when similar SST trends (and hence modes of variability) are considered.

Until this is better understood there are implications for future projections of impacts related to changes in surface humidity such as heat stress, food security and possibly extremes of the hydrological cycle.

**Acknowledgments**

We thank Adrian Simmons and an anonymous referee whose comments helped refine this manuscript, and also Gareth Jones, Fraser Lott, Nikolaos Christidis, Ben Booth and Rob Chadwick for interesting discussions during the course of this work. The authors were supported by the Joint BEIS/Defra Met Office Hadley Centre Climate Programme (GA01101).

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
