# Peer review of "Comparison of land-surface humidity between observations and CMIP5 models"

_Earth System Dynamics, 2017_

## Referee Comment (RC1) · Anonymous Referee #1 · 14 Mar 2017

The paper presents a comparison of climate model simulated humidity with that in a global observed dataset. As the authors mention, both specific and relative humidity are important quantities affecting health, productivity and resources. Therefore, the topic is important and relevant to the scientific community. However, there are some concerns, as I mention below, on the datasets and methods adopted in the paper that need to be addressed first. I am therefore recommending a major revision at this stage.

Firstly, unlike the abstract and introduction, the rest of the paper presents too much information (in many cases without stating the physical reasons or logically arising from the preceding section) that is hard to retain for the reader. Instead, one would expect concise sections which clearly explain in what aspects and why model simulations match with observations, where they do, and why they don't match with observations where they don't. For example, in Section 3: what causes the regional trends in tem-

perature which in turn influence humidity changes? While Section 4 states that water availability is not limited in the models, Section 6 states that the models are water limited to some extent: a contradiction?. Why do model simulations with greenhouse gases agree well with observations in the Northen Hemisphere (Section 4), and why do they don't agree for the Southern Hemisphere? Why does the T-q relationship in the models show better match with observations as compared to the T-rh? Moreover, all comparisons presented by the authors are either in the form of nature of trends or a plot of annual time series. What about the seasonal cycles and the spatial correlations?

Sections 4 and 6 each provide a helpful summary, while Section 5 does not have one. I am not sure how the addition of an atmosphere-only modeling experiment has added value to the analysis. Further, the authors present limited discussion on circulation patterns that play a major role in altering in drying or moistening of a region.

The observed dataset that the authors use are a single source global dataset that, as the authors themselves note, suffers from severe limitations including limited number of stations at several locations. How are the reported trends (or lack thereof) substantiated for such locations? And I wonder why satellite observations were not used in this study to augment the ground-based records. Moreover, for hydrometerological variables, considerable disagreements exist between different reanalyses products. How does the other reanalyses products compare with respect to ERA-I and HadISDH? How could differences therein, if any, be resolved to arrive at a robust observed record of humidity in the first place?

I think it is a little unfair to expect climate models to represent relative humidity, particularly at regional scales, given that the models are not equipped with relevant physical processes at such scales. Isn't it already well-known to the climate science community that the GCMs are not the best tools for hydrometeorologic variables, particularly relative humidity? Therefore, the regional, inter-decadal and physical differences in humidity that the authors report are generic and obvious. It would be much more compelling to see the comparison between large scale atmospheric variables that are

associated with humidity changes, in model simulations and observations so that the representation of such processes (and not humidity per se) may be improved in the next generation of climate models.

Further, since the authors are dealing with land-surface specific humidity and they suspect, perhaps correctly, that a major part of the unexplained differences are due to land-surface processes, would it not be logical to conduct a land-surface modeling experiment also, rather than that just using an atmosphere-only GCM?

The authors link the rising and falling trends in humidity with the so-called 'hiatus'. It is worth mentioning that rising and falling trends lie very much within the natural climate variability in presence of long-term persistence (see Koutsoyiannis and Montanari, WRR, 2007) and estimating trends based on 10 or 15 years of data is not a statistically robust exercise at all. Therefore, the decadal trend comparisons (Fig. 1 – 4) are not the best metrics to consider.

Other comments: Section 3: ii) discussions about projections may be deleted to avoid confusion. Section 3: iii) the Indian region has marked drying trend in the last half-century, contradictory to the 'moistening' reported by the authors. Page 1, line 15: check grammar. Page 2, line 1: 'Water vapor is the primary': suggest replacing 'the' with 'a'. Page 16, line 31: check grammar. Page 30, line 24: 2016 like to reach – we are already in 2017, therefore 2016 is not future anymore. Page 32, line 7: sentence is not clear.

---

## Referee Comment (RC2) · A.J. Simmons (Referee) · 16 Mar 2017

**Review of "Comparison of land-surface humidity between observations and CMIP5 models" by R.J.H. Dunn and co-authors**

This paper provides a welcome comparison between a hierarchy of datasets based on observations and models. The choice of datasets is a reasonable one, although a little more reference to other datasets could be given. I also find the level of discussion reasonable for a paper of this type. I thus consider the paper worthy of publication subject only to minor revisions to take into account the points listed (in no order of priority) below.

1. Page 2, lines 29 and 30. It would perhaps be better to add a word such as approximately, so the text reads "Relative humidity over oceans from reanalyses appears approximately constant ...". Whilst the reanalyses carried out to date probably do not assimilate moisture data over the oceans well enough to detect reliably a trend over the ocean, Hersbach *et al*. (2014; doi: 10.1002/qj.2528) do show a slight decline in dew-point depression in the ERA-20CM ensemble of model integrations with prescribed SST and CMIP5 forcings. Simmons *et al*. (2017; doi:10.1002/qj.2949) show a slight change over time in the difference between marine air temperature and SST in both the ERA-Interim and the JRA-55 analyses. This is a feature also of CMIP5 models (Cowtan *et al*., 2015; doi: 10.1002/2015GL064888). There may thus be a slight shift in the relative humidity of near-surface air over the oceans over time, even if it cannot be reliably detected directly in the reanalyses.

2. Page 3, lines 13 and 14. The reference here to thermal comfort of humans and livestock, and productivity, is rather repetitive of what is stated in the third paragraph on page 2.

3. Page 3, line27. The term *historicalNAT* should be explained here, where it first appears, rather than later.

4. Page 3, line 33. It would be better if the sentence referring to Sect 2 were to appear before the sentence referring to Sect 3, rather than before a sentence that begins "Finally ...".

5. Page 6, line 15 and 16. A stronger justification of the selection of ERA-Interim as the reanalysis to be included in this study could be given. Only ERA-Interim and JRA-55 of the major atmospheric reanalyses provide direct analyses of 2m temperature and humidity observations. Willett *et al*. (2016) shows much better agreement between ERA-Interim and JRA-55 than between either of these reanalyses and MERRA-2, especially for relative humidity. It probably would add little to include JRA-55 as well as ERA-Interim in this study, given that its main focus is on the comparison with CMIP5 models rather than the intercomparison of reanalyses, and the humidity analyses from ERA-Interim are the better documented in the literature. MERRA-2 cannot be recommended for use in this study as it does not give a reliable time series of two-metre temperature (Simmons *et al*., 2017; doi:10.1002/qj.2949). Its inhomogeneity is much larger than that which arises from the ERA-Interim SST changes.

6. Page 6, lines 24 to 28. It's more complicated than stated in that there was an ERA-Interim SST change in January 2002 as well as one in June 2001. It was the combined effect of these two changes that shifted SST about 0.1K colder, a shift we now adjust for in studies such as Simmons *et al*. (2017). The other changes in source of SST analysis are more minor in their impact. Apologies for more self-citation, but reference could be given to the summary of the SST and sea-ice changes given by

Simmons and Poli (2015; doi: 10.1002/qj.2422), which notes the January 2002 change as well as the June 2001 one.

7. Page 6, line 29. "inhomogeneities" might be a better word than "instabilities".

8. Page 8. The three columns of panels in Figure 1 could be headed "Air temperature", "Specific Humidity" and "Relative Humidity" to help someone glancing through the paper. At present these words appear only in the headings of the lowermost panels.

9. Page 12, line 1. m/m-1. is zero under the usual convention of doing the divide before the minus. Do the authors mean m/(m-1.)?

10. Page 12, line 19. Whilst the CMIP5 models all have larger positive trends than observations indicate for the period 1996-2015, the observed warmth of 2016 (and perhaps 2017, given it has started warm and another El Nino is forecast) makes one wonder what the conclusions will be when we have CMIP6 integrations – even if the models do not change much. Some discussion relevant to this is given in section 7 (see also comment 20) and perhaps a reference to the discussion later in the paper could be given on page 12.

11. Page 14, line 2. Missing word "be" before "expected".

12. Page14, line 17. Reference here could be made to Hersbach *et al*. (2014), as the finding for HadGEM3-A mirrors that already made for ERA-20CM. Hersbach *et al*. showed that in model runs for the period 1900-2010, the driest conditions occurred in the final decade, both for surface air humidity and for soil moisture. Although the dryness of the ensemble mean was not as large as that in ERA-Interim, a few of the ten ensemble members reached levels of dryness similar to those reached by ERA-Interim.

13. Page 17. The final sentence of the figure caption states that all climatologies were calculated over the 1975-2009 period. This cannot be the case for ERA-Interim.

14. Page 21, line 27. Missing word "in" before "ingested".

15. Page 21, line 27. The change in January 2002 as well as June 2001 is relevant in this regard, as discussed in comment 6.

16. Page 21, line 32. I baulked a little when I read of relative humidity becoming more arid. Is this a correct use of the word "arid"? Land and air can become more arid, but relative humidity?

17. Page 24, page 26 and page 27. The colours referred to in these figure captions are not all correct. The captions state that ERA-Interim is shown in blue, whereas I think from the figure that the colour is deep pink (or red). The captions of Figures 8 and 10 refer to a yellow that should be orange, and a red that should be blue.

18. Page 28, line 10. "ERA-interim" should be "ERA-Interim".

19. Page 29, line 28. "spare" should be "sparse".

20. Page 30, lines 15 to 25. A more nuanced discussion should be given, and should reflect that we now have the figures for 2016. Reference could be given to Simmons *et al.* (2017), who compare the temperatures from several datasets for the period up to July 2016, with numbers until the end of 2016 presented at

http://climate.copernicus.eu/resources/data-analysis/average-surface-air-temperature-analysis

The datasets show some variability when it comes to how much warmer 2016 was than 2015, with HadCRUT4 showing little difference, and datasets that provide values over more of or all the Arctic and Antarctic giving warmer values in 2016 than 2015, the difference approaching an unusually large 0.2°C in the case of ERA-Interim and JRA-55. However, these differences are probably of limited relevance in the context of the paper under review, as they stem mainly from regions over or near to sea-ice, which has been of record low extent in both the Arctic and the Antarctic in recent months.

Simmons *et al.* (2017) is also a relevant reference in that the paper shows that the latest estimates of the trend from 1998-2012 are (for all datasets examined) higher than the central estimate made in IPCC AR5, suggesting less of a "hiatus" or slowdown in warming than first indicated.

21. Page 31, line 20. I am puzzled by the sentence that the reanalyses do not show as clear a warming trend as the observations. This is not what is concluded by Simmons *et al.* (2017) when comparing ERA-Interim (adjusting for its warmer SSTs prior to 2002) and JRA-55 with GISTEMP, HadCRUT4 and NOAAGlobalTemp.

22. page 32, line 4. The text here should refer to "the drying identified in ERA-Interim and HadISDH". It should be in this order as the drying was identified first in ERA-Interim (published in 2010, confirmed then by an intermediate Hadley Centre dataset) and confirmed later by HadISDH (published in 2014).

Adrian Simmons
16 March 2017

---

## Author Comment (AC1) · 25 May 2017

*We thank Adrian for his time in commenting on this manuscript. We respond to each point below with our responses in italics.*

Review of "Comparison of land-surface humidity between observations and CMIP5 models" by R.J.H. Dunn and co-authors

This paper provides a welcome comparison between a hierarchy of datasets based on observations and models. The choice of datasets is a reasonable one, although a little more reference to other datasets could be given. I also find the level of discussion reasonable for a paper of this type. I thus consider the paper worthy of publication subject only to minor revisions to take into account the points listed (in no order of priority) below.

1. Page 2, lines 29 and 30. It would perhaps be better to add a word such as approximately, so the text reads "Relative humidity over oceans from reanalyses appears approximately constant ...". Whilst the reanalyses carried out to date probably do not assimilate moisture data over the oceans well enough to detect reliably a trend over the ocean, Hersbach et al. (2014; doi: 10.1002/qj.2528) do show a slight decline in dew-point depression in the ERA-20CM ensemble of model integrations with prescribed SST and CMIP5 forcings. Simmons et al. (2017; doi:10.1002/qj.2949) show a slight change over time in the difference between marine air temperature and SST in both the ERA-Interim and the JRA-55 analyses. This is a feature also of CMIP5 models (Cowtan et al., 2015; doi: 10.1002/2015GL064888). There may thus be a slight shift in the relative humidity of near-surface air over the oceans over time, even if it cannot be reliably detected directly in the reanalyses.

*Response: We have added the "approximately" as suggested and have added extra sentences to capture the arguments outlined in this suggestion along with the appropriate references.*

2. Page 3, lines 13 and 14. The reference here to thermal comfort of humans and livestock, and productivity, is rather repetitive of what is stated in the third paragraph on page 2.

*Response: We have removed this clause and refereed to the earlier text.*

3. Page 3, line27. The term historicalNAT should be explained here, where it first appears, rather than later.

*Response: We have added a quick explanation at this point in the text and referred forward to the more detailed explanation in Section 2.2*

4. Page 3, line 33. It would be better if the sentence referring to Sect 2 were to appear before the sentence referring to Sect 3, rather than before a sentence that begins "Finally ...".

*Response: We have moved this sentence to the beginning of the previous paragraph and merged the remaining sentence "Finally..." in as well.*

5. Page 6, line 15 and 16. A stronger justification of the selection of ERA-Interim as the reanalysis to be included in this study could be given. Only ERA-Interim and JRA-55 of the major atmospheric reanalyses provide direct analyses of 2m temperature and humidity observations. Willett et al. (2016) shows much better agreement between ERA-Interim and JRA-55 than between either of these reanalyses and MERRA-2, especially for relative humidity. It probably would add little to include JRA-55 as well as ERA-Interim in this study, given that its main focus is on the comparison with CMIP5 models rather than the intercomparison of reanalyses, and the humidity analyses from ERA-Interim are the better documented in the literature. MERRA-2 cannot be recommended for use in this study as it does not give a reliable time series of two-metre temperature (Simmons et al., 2017; doi:10.1002/qj.2949). Its inhomogeneity is much larger than

that which arises from the ERA-Interim SST changes.

*Response: We have added discussion around the different reanalyses products, as outlined in this comment. We agree that adding JRA-55 may not add anything to the comparison assessment, but it would be good to do this for completeness. However, given the global comparisons of reanalyses products are performed in the annual BAMS State of the Climate, and that the focus of this manuscript is on the CMIP5 model ensemble (as you note), we have not added this product.*

6. Page 6, lines 24 to 28. It's more complicated than stated in that there was an ERA-Interim SST change in January 2002 as well as one in June 2001. It was the combined effect of these two changes that shifted SST about 0.1K colder, a shift we now adjust for in studies such as Simmons et al. (2017). The other changes in source of SST analysis are more minor in their impact. Apologies for more self-citation, but reference could be given to the summary of the SST and sea-ice changes given by Simmons and Poli (2015; doi: 10.1002/qj.2422), which notes the January 2002 change as well as the June 2001 one.

*Response: We have expanded this discussion in light of the information kindly given and added the two citations as well.*

7. Page 6, line 29. "inhomogeneities" might be a better word than "instabilities".

*Response: We have replaced the word as suggested.*

8. Page 8. The three columns of panels in Figure 1 could be headed "Air temperature", "Specific Humidity" and "Relative Humidity" to help someone glancing through the paper. At present these words appear only in the headings of the lowermost panels.

*Response: Headings added*

9. Page 12, line 1. m/m-1. is zero under the usual convention of doing the divide before the minus. Do the authors mean m/(m-1.)?

*Response: We do - thank you for spotting this. Formula updated*

10. Page 12, line 19. Whilst the CMIP5 models all have larger positive trends than observations indicate for the period 1996-2015, the observed warmth of 2016 (and perhaps 2017, given it has started warm and another El Nino is forecast) makes one wonder what the conclusions will be when we have CMIP6 integrations – even if the models do not change much. Some discussion relevant to this is given in section 7 (see also comment 20) and perhaps a reference to the discussion later in the paper could be given on page 12.

*Response: We have added a reference forward to Section 7 at this point. We acknowledge that the next few years observations, both temperature and humidity, may result in this work being reassessed.*

11. Page 14, line 2. Missing word "be" before "expected".

*Response: Added*

12. Page14, line 17. Reference here could be made to Hersbach et al. (2014), as the finding for HadGEM3-A mirrors that already made for ERA-20CM. Hersbach et al. showed that in model runs for the period 1900-2010, the driest conditions occurred in the final decade, both for surface air

humidity and for soil moisture. Although the dryness of the ensemble mean was not as large as that in ERA-Interim, a few of the ten ensemble members reached levels of dryness similar to those reached by ERA-Interim.

*Response: Thank you - we have added a reference and note about these results at the end of this paragraph.*

13. Page 17. The final sentence of the figure caption states that all climatologies were calculated over the 1975-2009 period. This cannot be the case for ERA-Interim.

*Response: We have updated the caption to reflect the different temporal coverage of ERA-Interim*

14. Page 21, line 27. Missing word "in" before "ingested".

*Response: Added*

15. Page 21, line 27. The change in January 2002 as well as June 2001 is relevant in this regard, as discussed in comment 6.

*Response: We have referred back to the discussion of both of these changes in Section 2.2 and reduced the detail at this point in the manuscript.*

16. Page 21, line 32. I baulked a little when I read of relative humidity becoming more arid. Is this a correct use of the word "arid"? Land and air can become more arid, but relative humidity?

*Response: We have replaced this with "less saturated" to ensure clarity.*

17. Page 24, page 26 and page 27. The colours referred to in these figure captions are not all correct. The captions state that ERA-Interim is shown in blue, whereas I think from the figure that the colour is deep pink (or red). The captions of Figures 8 and 10 refer to a yellow that should be orange, and a red that should be blue.

*Response: Our apologies that these captions had not been updated to the final colours used. These have now been amended. Those in the supplement were correct.*

18. Page 28, line 10. "ERA-interim" should be "ERA-Interim".

*Response: Amended*

19. Page 29, line 28. "spare" should be "sparse".

*Response: Amended*

20. Page 30, lines 15 to 25. A more nuanced discussion should be given, and should reflect that we now have the figures for 2016. Reference could be given to Simmons et al. (2017), who compare the temperatures from several datasets for the period up to July 2016, with numbers until the end of 2016 presented at http://climate.copernicus.eu/resources/data-analysis/average-surface-air-temperature-analysis
The datasets show some variability when it comes to how much warmer 2016 was than 2015, with HadCRUT4 showing little difference, and datasets that provide values over more of or all the Arctic and Antarctic giving warmer values in 2016 than 2015, the difference approaching an unusually large 0.2°C in the case of ERA-Interim and JRA-55. However, these differences are probably of

limited relevance in the context of the paper under review, as they stem mainly from regions over or near to sea-ice, which has been of record low extent in both the Arctic and the Antarctic in recent months.

Simmons et al. (2017) is also a relevant reference in that the paper shows that the latest estimates of the trend from 1998-2012 are (for all datasets examined) higher than the central estimate made in IPCC AR5, suggesting less of a "hiatus" or slowdown in warming than first indicated.

*Response: This was also noted by the other reviewer, and we have updated this section with the data from 2016 (which wasn't available when preparing for submission). Thank you for the suggestions to include Simmons et al. (2017) which we have done as well as the extra discussion on polar coverage in the datasets.*

21. Page 31, line 20. I am puzzled by the sentence that the reanalyses do not show as clear a warming trend as the observations. This is not what is concluded by Simmons et al. (2017) when comparing ERA-Interim (adjusting for its warmer SSTs prior to 2002) and JRA-55 with GISTEMP, HadCRUT4 and NOAAGlobalTemp.

*Response: We have removed that clause from the final sentence of that paragraph.*

22. page 32, line 4. The text here should refer to "the drying identified in ERA-Interim and HadISDH". It should be in this order as the drying was identified first in ERA-Interim (published in 2010, confirmed then by an intermediate Hadley Centre dataset) and confirmed later by HadISDH (published in 2014).

*Response: Text amended to include reference to ERA-Interim*

---

## Author Comment (AC2) · 25 May 2017

*We thank the reviewer for their time in commenting on this manuscript. We respond to each point below with our responses in italics.*

The paper presents a comparison of climate model simulated humidity with that in a global observed dataset. As the authors mention, both specific and relative humidity are important quantities affecting health, productivity and resources. Therefore, the topic is important and relevant to the scientific community. However, there are some concerns, as I mention below, on the datasets and methods adopted in the paper that need to be addressed first. I am therefore recommending a major revision at this stage.

Firstly, unlike the abstract and introduction, the rest of the paper presents too much information (in many cases without stating the physical reasons or logically arising from the preceding section) that is hard to retain for the reader. Instead, one would expect concise sections which clearly explain in what aspects and why model simulations match with observations, where they do, and why they don't match with observations where they don't. For example, in Section 3: what causes the regional trends in temperature which in turn influence humidity changes?

*Response: We note that this paper is long, which is a result of the multiple variables (air temperature, relative and specific humidity), the nature of the comparisons (temporal, spatial, spatio-temporal and inter-variable) do take time to got through, especially if including regional detail. Our attempt was to present the different comparisons separately and then bring these together during the discussions. We realise that this results in sections which stand alone, without building on the previous one, but they are parallel investigations into the modelled behaviour of the humidity variables In the final paragraph of the introduction we outline the structure of the paper and how we present the analysis done therein. We have ensured that each section is re-introduced to indicate what is being presented and assessed in each one, and hope that these set the scene for readers.*

*The reason for doing this assessment was not to perform an attribution analysis, however it is to describe the data set in the context of climate models – although an eye is kept on the potential for attribution throughout. We did not intend to present many of the answers to "why" questions, but focus more on the "how" and "where" models and observations differ when presented in a number of ways. Adding formal assessments to the "why" questions would add further to an already lengthy paper.*

*As a response to your example of what causes regional trends in temperature, we know of no clear answer at present. Jones et al (2013) analyse spatial temperature trends over 1979-2010 shown in Figs 8 and 9. Cooling compared to historical and historicalGHG models is seen in the Pacific and Southern Oceans. The effects in the Pacific may be the result of the PDO as several CMIP5 members look similar so there is some representation and hence need not be model error.*

*The differences in the Southern Ocean may be from the Southern Annual Mode plus some forcing contribution (Jones et al reference Trenberth et al, 2007 and Karpechko et al, 2009) and again several simulations show similar cooling here (see Fig 9 bottom panel) suggesting that it need not be model error. A further complication is whether low cloud cover plays a role, as this can be associated with aerosols, and Jones et al (2013) show better agreement with global temperature trends when using models which include both direct and indirect aerosol effects. We have added a section outlining these arguments into the summary for the temporal assessment as well as the discussion along with the appropriate references.*

*As most historical members do not show these patterns (as hence reflected in warmer ensemble*

*means) it is possible that the coupled models are biased not enough decadal-scale variability, which is generally thought to be the case. However, we don't know whether the world we live in has been particularly strange (regardless of any anthropogenic influences) over the recent decades.*

*Jones, G. S., P. A. Stott, and N. Christidis (2013), Attribution of observed historical near–surface temperature variations to anthropogenic and natural causes using CMIP5 simulations, J. Geophys. Res. Atmos., 118, 4001–4024, doi:10.1002/jgrd.50239.*

*Karpechko, A. Y., N. P. Gillett, G. J. Marshall, and J. A. Screen (2009), Climate impacts of the Southern Annular Mode simulated by the CMIP3 models, J. Climate, 22, 3751–3768.*

*Trenberth, K. E., et al. (2007), Observations: Surface and atmospheric climate change, in Climate Change 2007: The Physical Science Basis. Contribution of Working Group I to the Fourth Assessment Report of the Intergovernmental Panel on Climate Change, edited by S. Solomon, et al., pp. 235–336, Cambridge University Press, Cambridge, United Kingdom and New York, NY, USA.*

While Section 4 states that water availability is not limited in the models, Section 6 states that the models are water limited to some extent: a contradiction?

*Response: To clarify, in Section 4 we state that the "behaviour of the observed and modelled humidity since 1973 suggests that water availability is less of a limiting factor in the models than the observations". In Section 6, we state that "the larger slopes and higher correlations in the CMIP5 models suggests that there could be less water availability limitation compared to the observations, especially in the Southern Hemisphere". Both these statements suggest that water availability may be less limited in the CMIP5 models than in the observations. Our intention was not to suggest that there are no limitations on the water availability in Section 4 or something to the contrary in Section 6.*

*We have slightly changed the wording in Section 6 to match that from Section 4 to try and prevent this confusion.*

*The remaining parts mentioning water availability in this work do so in a general way rather than comparing models and observations, so these have not been amended.*

Why do model simulations with greenhouse gases agree well with observations in the Northen Hemisphere (Section 4), and why do they don't agree for the Southern Hemisphere?

*Response: We presume that you are referring to both the historical and the historicalGHG experiments with this comment. As noted in our response above, there is cooling in the Southern Ocean, which of course dominates the Southern Hemisphere surface. And hence this may be reflected in the land-based temperature data of HadISDH. In Jones et al. (2013), Figure 8 shows that the ensemble mean of the Southern Hemisphere historicalGHG experiments are very similar to the historical, but these two differ from the historicalNat experiments and observations. A cooler ocean would result in less moisture present in the atmosphere, and then advected over land; hence a lower specific humidity for the observations and historicalNat compared to the historical and historicalGHG experiments (as is seen in the Supplementary Information Fig 27). As the land surface in the Southern Hemisphere is not warming as fast as in the Northern (and also we note the data coverage limitations) there is no clear difference in the response for the relative humidity as a result. As noted above, we have added sections into the paper to outline these arguments.*

Why does the T-q relationship in the models show better match with observations as compared to the T-rh?

*Response: We do not have a complete explanation for this. It may be partly the result of relative humidity being a more sensitive variable than specific humidity. This is something that we have purposefully left for further work. As noted in previous responses, the aim of this paper was not to uncover the causes of the differences between the observed and modelled responses of temperature and humidity, but to present their existence through a number of analyses.*

*Another aspect to note (which we now have done in the manuscript) is that we show each experiment separately rather than ensemble means (which would smooth out some of the inter-annual variability, and so reduce the power of this assessment). The observations are, by their nature, a single realisation, and hence this contrast may act to indicate an over-correlation.*

*It is expected that the T-q relationship would be tighter than the T-rh one, so the scatter in the latter is larger – but also then the scatter for the observations will be larger. As the observations are only a single realisation being compared to many realisations for each model (and experiment) then this could give a false impression as to the level of (dis)agreement.*

Moreover, all comparisons presented by the authors are either in the form of nature of trends or a plot of annual time series. What about the seasonal cycles and the spatial correlations?

*Response: We now note that we have not assessed any seasonal patterns in the summary to Section 4 or spatial correlations in the new summary to Section 5 (see comment and response below). We agree that there are limitations to annual timeseries and linear trends (especially the latter), however feel that these simple measures are appropriate for this initial assessment of the differences between model and observed values. Our focus was also initially on the longer-term behaviour, and so showing monthly (or seasonal) timeseries would hide any signal in the higher variability at these shorter timescales.*

*We have now included monthly global timeseries plots in the Supplementary Material to show that (1) the trends found are of similar relative magnitudes to those from annual data and (2) that the models show clear monthly variability for temperature and specific humidity (even if the magnitudes do not match for individual members) but there no obvious signal for monthly variability in relative humidity. We have added this into the summary of Section 4.*

Sections 4 and 6 each provide a helpful summary, while Section 5 does not have one.

*Response: We have added a short summary (mindful of the comments above) for Section 5 and also moved the final two paragraphs of this section into this new summary where they probably now sit best.*

I am not sure how the addition of an atmosphere-only modeling experiment has added value to the analysis.

*Response: The intention of including an AMIP model was to see whether using prescribed sea-surface temperatures would improve the way the land-ocean heating contrast appeared in the models, and hence, the behaviour of the humidity measures. An AMIP model will also explicitly remove any erroneous ocean-atmosphere coupling, which would impact the SST patterns and related moisture content of the atmosphere, though of course at the expense of introducing a parameterisation in the model atmosphere. An important aspect of including AMIP experiments is that this model would also ensure that the modes of variability of large-scale oscillations (e.g. ENSO, NAO) were also matched to observations. As we are sure you are aware, in the coupled models, the phases of modes of decadal scale variability will be utterly scrambled. By using the*

*atmosphere-only model, we have attempted to remove a possibly large component which could be confounded with anthropogenic influences (see also our response to your comment about large scale circulation patterns, below). Using the observed SSTs, should mean that the oceans do not heat as much as the land surface, and so restricting the amount of water vapour in the atmosphere (relatively constant specific humidity) and hence a decline of relative humidity over land. We have expanded on this in the updated text to clarify why this model has been included in the assessment and referred forward to the discussion section where this is also presented.*

Further, the authors present limited discussion on circulation patterns that play a major role in altering in drying or moistening of a region.

*Response: We have only briefly touched on the fact that changes in circulation patterns would affect the humidity measures of a region. As outlined above, our intention was to highlight the current difference between CMIP5 models and observations (on temporal, spatial and spatio-temporal lines) along with some initial investigations as to possible causes. We have noted that a more in depth analysis of the effects of circulation changes is again something for a future, targeted study.*

*However, the effect of circulation changes is something which using an atmosphere-only model starts to address. As the ocean is a driver for a number of scale circulation patterns, using prescribed sea-surface temperatures should improve their representation, and any changes in the temperature and humidity measures similarly, especially at short timescales. We state this in the temporal discussion for specific humidity (Section 4.2). We have now added this thread to the discussion section as well.*

*We do note that not all circulation patterns are linked to ocean temperatures, and hence it is unlikely that the representation of all these within the atmosphere-only model is improved.*

The observed dataset that the authors use are a single source global dataset that, as the authors themselves note, suffers from severe limitations including limited number of stations at several locations. How are the reported trends (or lack thereof) substantiated for such locations? And I wonder why satellite observations were not used in this study to augment the ground-based records.

*Response: We feel that the comment of "severe" is a little harsh, but overall there will always be issues with coverage of observational data until data sharing occurs more freely. In the creation of HadISDH, care has been taken to ensure that the grid box averages presented are fair representations of the underlying station data. Of course, in areas with high network densities, this is easier than those in low. This is why this dataset includes uncertainty estimates along with the central values.*

*When calculating grid box trends, we have ensured that the completeness needs to be high (80% of months present) to reduce the effect of poor temporal coverage. We also use a trend fitting algorithm (median of pairwise slopes) which is robust to outliers. We note that the uncertainty information available with HadISDH has not been included in this trend fitting process. These trends are also shown in Willett et al, 2015, where the trends that are considered to be significantly different from a zero trend – where the 5th and 95th percentiles of the pairwise slopes are in the same direction - are highlighted.*

*To give an indication of the regions which have small/large numbers of stations in a grid box, on Figure 1, we have highlighted the boxes with more than 3 stations in them with a thicker outline in all of the panels.*

*The merging of in-situ and satellite based data is a non-trival operation. As with the in-situ observations presented here, satellite measurements have their own (different) problems in creating unbiased, global fields. Satellites can obtain lower, mid- and upper tropospheric humidity values, but the link between the lower-tropospheric and near-surface humidities isn't clear [CHECK]. Creating a merged in-situ/satellite humidity product would be a substantial work in itself, although we agree that if this were available, then it would be very useful in an assessment similar to that presented here.*

Moreover, for hydrometerological variables, considerable disagreements exist between different reanalyses products. How does the other reanalyses products compare with respect to ERA-I and HadISDH? How could differences therein, if any, be resolved to arrive at a robust observed record of humidity in the first place?

*Response: The other reviewer also noted that we have only included one reanalysis product. There are three up-to-date products that could be included: ERA-Interim, JRA-55 and MERRA-2. These are all included in the annual BAMS State of the Climate report, showing the global average timeseries for humidity (see Willett, K. M., Berry, D. I., Bosilovich, M. G., and Simmons, A.: Surface humidity, in BAMS "State of the Climate in 2015", 2016). We have added a discussion of the other reanalysis products in light of the comments from Adrian Simmons in this section. The summary of these is that there are still issues with MERRA-2 for the most recent years, and it does not give a reliable timeseries for 2m air temperature. Both ERA-Interim and JRA-55 agree well, and both agree equally badly with MERRA-2. As our main focus in this work is on the comparisons of the observations with CMIP5 models, rather than with different reanalysis products, and the close match between ERA-Interim, HadISDH and JRA-55 shown in the State of the Climate reports, adding JRA-55 may not add much to this work.*

*An assessment of the differences shown for hydrometeorological variables across all reanalysis products is beyond the scope of this work. Although reanalyses do provide complete global coverage, they are not perfect reconstructions of the state of the atmosphere and oceans over their period of record. Indeed, we note some of the changes in ERA-Interim over time which result in inhomogeneities in the timeseries. Also, comparing the lower row of Figures 5 and 6 show that there are regional differences in both the climatology and trend behaviour of ERA-Interim and HadISDH. The best way of arriving at a robust observed humidity record is to increase the amount of in-situ observations available over the last century, and fill in the spatial and temporal gaps currently present in HadISDH. These can also then be assimilated into the reanalyses products, improving their representation of the atmosphere.*

*By showing the differences, temporally, spatially and spatio-temporally, for humidity in observation and reanalyses products, issues within both the observed data and the reanalyses (as well as the models) are made clear to the community in this manuscript. To determine what improvements are needed in the reanalysis processes is beyond the scope of this work and the expertise of the authors. However, the comparisons do highlight where further work is required in the observational record, and we, along with colleagues, are working to improve the number of observations available to the community through projects such as ACRE (Allan et al, 2011) and new Copernicus Climate Change Services.*

*Allan, R., Brohan, P., Compo, G.P., Stone, R., Luterbacher, J. and Brönnimann, S., 2011. The international atmospheric circulation reconstructions over the earth (ACRE) initiative. Bulletin of the American Meteorological Society, 92(11), pp.1421-1425.*

I think it is a little unfair to expect climate models to represent relative humidity, particularly at regional scales, given that the models are not equipped with relevant physical

processes at such scales. Isn't it already well-known to the climate science community that the GCMs are not the best tools for hydrometeorologic variables, particularly relative humidity? Therefore, the regional, inter-decadal and physical differences in humidity that the authors report are generic and obvious. It would be much more compelling to see the comparison between large scale atmospheric variables that are associated with humidity changes, in model simulations and observations so that the representation of such processes (and not humidity per se) may be improved in the next generation of climate models.

*Response: We do not believe that it is well known that hydrometeorological variables are not well captured by GCMs. It is our understanding that the observed changes in relative humidity, although having a number of hypothesised causes, are not definitively understood. And so that observed changes in these variables are awaiting understanding in terms of variability which can only be assessed through explorations using physical modelling. We think that the comparisons performed in this work are an essential first step to assessing the potential for and understanding the results of future, further attribution assessments. We do not think that differences between the models (especially ensemble means) and the observed data are can be unequivocally put down to model error nor to inherently high variability and sensitivity which only modelling assessments can reveal.*

Further, since the authors are dealing with land-surface specific humidity and they suspect, perhaps correctly, that a major part of the unexplained differences are due to land-surface processes, would it not be logical to conduct a land-surface modeling experiment also, rather than that just using an atmosphere-only GCM?

*Response: We agree that a number of further experiments could be run to determine the eventual cause of the differences between the observed humidity measures and the CMIP5 experiments assessed here. Our inclusion of the AO-GCM was to see whether prescribed SSTs would improve the land-ocean heating contrast representation in the models, and hence whether the humidity behaviour would match the observations more closely (see response above). A first step beyond the "vanilla" CMIP5 experiments, if you will.*

*The aim of this work was to note and highlight the level of agreement (or not) between the CMIP5 historical (including historicalNat and historicalGHG) experiments and the observed land humidity measures. Even by just restricting to the assessments currently in the manuscript, without including the seasonal and spatial correlations suggested above, this is already a lengthy document (we have tried to condense where possible and include the structural suggestions you note above). A number of avenues for further investigation present themselves, including land-surface modelling.*

*Given the length of this manuscript (as noted by you above), we suggest that including this experiment along with the other additions would make the manuscript rather unwieldy and even longer. We now note more clearly at the end of the manuscript, possible avenues for future work, including the land-surface experiments suggested above, and think that the resulting manuscripts could then be more easily targeted at specific aspects of this issue.*

The authors link the rising and falling trends in humidity with the so-called 'hiatus'. It is worth mentioning that rising and falling trends lie very much within the natural climate variability in presence of long-term persistence (see Koutsoyiannis and Montanari, WRR, 2007) and estimating trends based on 10 or 15 years of data is not a statistically robust exercise at all. Therefore, the decadal trend comparisons (Fig. 1 – 4) are not the best metrics to consider.

*Response: We acknowledge that using trends on short periods of data is not a robust method of assessing changes over time. Although the time-span of the dataset is just over 40 years (so giving two periods of 20year trends in Figs 1-4), we also use a trend fitting method which is insensitive to outliers (median of pairwise slopes) and also have tried to only indicate differences from no trend. In this work we are not trying to understand observed changes in terms of linear responses modulated with a random noise component (for which, of course these 20 years of data would be shorter than necessary). If, however, the shorter period changes are related to real events, then using shorter sections of time series is necessary in the quest to understand them.*

*We thought it appropriate to make links with the so-called "hiatus" (which we note here, and in the manuscript itself, is no longer as apparent in the latest data releases) given the temporal alignment between the apparent slow-down in global surface temperature rise and the change in behaviour of the humidity measures. We note (and have now updated in light of your comment on the 2016 figures below) that the most recent annual global temperature values are the hottest on record, and that "it is expected that short period trends vary around trend measured over a longer period" in the final sentence of this paragraph.*

*We have rephrased the last few sentences to clarify these arguments and added the reference you suggest to allow readers to investigate this issues further more easily.*

Other comments:

Section 3: ii) discussions about projections may be deleted to avoid confusion.

*Response: We would prefer to keep these comments in as they illustrate that long-term changes in humidity measures are indicated by long model runs. However, we have re-ordered this paragraph and clarified it as well to ensure that this discussion does not cause confusion*

Section 3: iii) the Indian region has marked drying trend in the last half-century, contradictory to the 'moistening' reported by the authors.

*Response: In Fig 6, to which this section refers, panels (b) and (c) show green over the Indian subcontinent, linking to increased specific and relative humidity over the period of this dataset (1973-2015). Figure 1 shows decadal trends, and although the strongest moistening/saturation trends are in the earlier decades, a reversal is not presented in these figures. We acknowledge that this dataset does not cover the complete last half-century and so that changes in the 1950s and 60s are not presented here which may dominate when assessing over that scale.*

Page 1, line 15: check grammar.

*Response: Checked and sentence clarified*

Page 2, line 1: 'Water vapor is the primary': suggest replacing 'the' with 'a'.

*Response: Replaced*

Page 16, line 31: check grammar.

*Response: Checked and clarified*

Page 30, line 24: 2016 like to reach – we are already in 2017, therefore 2016 is not future anymore.

*Response: Not all the final assessments for 2016 were not available when preparing this draft. We have updated the sentence in light of the now-released figures.*

Page 32, line 7: sentence is not clear.

*Response: Sentence clarified*